# The $CO_2$ integral emission by the megacity of St. Petersburg as quantified from ground-based FTIR measurements combined with dispersion modelling

Dmitry V. Ionov[1], Maria V. Makarova[1], Frank Hase[2], Stefani C. Foka[1], Vladimir S. Kostsov[1], Carlos Alberti[2], Thomas Blumenstock[2], Thorsten Warneke[3], Yana A. Virolainen[1]

[1] Department of Atmospheric Physics, Faculty of Physics, St. Petersburg State University, Russia

[2] Karlsruhe Institute of Technology (KIT), Institute of Meteorology and Climate Research (IMK-ASF), Karlsruhe, Germany

[3] University of Bremen, Germany

*Correspondence to:* Dmitry V. Ionov (d.ionov@spbu.ru), Frank Hase (Frank.Hase@kit.edu) and Maria V. Makarova (m.makarova@spbu.ru)

**Abstract.** The anthropogenic impact is a major factor of the climate change which is highest in industrial regions and modern megacities. Megacities are a significant source of emissions of various substances into the atmosphere, including $CO_2$ which is the most important anthropogenic greenhouse gas. In 2019 and 2020, the mobile experiment EMME (Emission Monitoring Mobile Experiment) was carried out on the territory of St. Petersburg which is the second largest industrial city in Russia with a population of more than 5 million people. In 2020, several measurement data sets were obtained during the lockdown period caused by the COVID-19 (COronaVIrus Disease of 2019) pandemic. One of the goals of EMME was to evaluate the $CO_2$ emission from the St. Petersburg agglomeration. Previously, the $CO_2$ area flux has been obtained from the data of the EMME-2019 experiment using the mass balance approach. The value of the $CO_2$ area flux for St. Petersburg has been estimated as $89\pm28$ kt km$^{-2}$ yr$^{-1}$ which is three times higher than the corresponding value reported in the official municipal inventory. The present study is focused on the derivation of the integral $CO_2$ emission from St. Petersburg by coupling the results of the EMME observational campaigns of 2019 and 2020 and the HYSPLIT (HYbrid Single-Particle Lagrangian Integrated Trajectories) model. The ODIAC (Open-source Data Inventory for Anthropogenic $CO_2$) database is used as the source of the a priori information on the $CO_2$ emissions for the territory of St. Petersburg. The most important finding of the present study based on the analysis of two observational campaigns is a significantly higher $CO_2$ emission from the megacity of St. Petersburg as compared to the data of municipal inventory: $\sim$75800$\pm$5400 kt yr$^{-1}$ for 2019, $\sim$68400$\pm$7100 kt yr$^{-1}$ for 2020 versus $\sim$30000 kt yr$^{-1}$ reported by official inventory. The comparison of the $CO_2$ emissions obtained during the COVID-19 lockdown period in 2020 to the results obtained during the same period of 2019 demonstrated the decrease in emission of 10% or 7400 kt yr$^{-1}$.

**Keywords:** ground-based remote sensing, portable spectrometers, FTIR spectroscopy, mobile experiments, anthropogenic emissions in megacities, transport modelling of air pollutants, $CO_2$, ODIAC, HYSPLIT

# 1 Introduction

Accurate quantitative assessment of anthropogenic emissions into the atmosphere is necessary for studying the mechanisms and factors that determine the impact of changes in atmospheric composition on climate, ecosystems and human health. Also, such an assessment is important for the development and control of compliance of the national policies in the field of environmental and climate protection to international agreements, regulations and standards (Pacala et al., 2010; Ciais et al., 2015; UNFCCC, 2015). In 2018, World Meteorological Organisation (WMO) established the IG3IS division (Integrated Global Greenhouse Gas Information System). Its activities are related to international efforts relevant to the implementation of the Paris Agreement under the United Nations Framework Convention on Climate Change (UNFCCC, 2015). The main goal of IG3IS is "to expand the observational capacity for greenhouse gases (GHG), extend it to the regional and urban domains, and develop the information systems and modelling frameworks to provide information about GHG emissions to society" (IG3IS, 2020).

According to statistics for 2018 (UN, 2021), 4.2 billion people or about 55% of the World's population live in cities. Urban areas are responsible for more than 70% of global energy-related $CO_2$ emissions (Canadell et al., 2010). The vast majority of anthropogenic $CO_2$ emissions in developed countries are associated with the burning of fossil fuels (FF) and can be estimated with good accuracy on the basis of the total fuel consumption. At the same time, available data on regional and local emissions have a significantly lower level of confidence (Ciais et al., 2015; Bréon et al., 2015; Kuhlmann et al., 2019). Usually, to check the accuracy of the $CO_2$ emission inventories (the so-called "bottom-up" data), the independent "top-down" approach is applied which is based on a combination of atmospheric observations and numerical simulations. Currently, the efforts in this direction are being made by international scientific communities in the framework of such large-scale projects as, for example, the VERIFY project (https://verify.lsce.ipsl.fr/) and the $CO_2$ Human Emissions (CHE) project (https://www.che-project.eu/). As an example of successful implementation of the "top-down" approach one can mention the experience of the United Kingdom in the evaluation of greenhouse gas emission national inventory (Stanley et al., 2018; WMO Greenhouse Gas Bulletin, 2018). Disaggregation of national FF $CO_2$ emission estimates provided the possibility to compile ODIAC (Open-source Data Inventory for Anthropogenic $CO_2$) which is a high resolution global open database of anthropogenic $CO_2$ emissions (Oda and Maksyutov, 2011; Oda et al. 2018).

Recently, much attention has been paid to the improvement of the estimates of the $CO_2$ emissions by the world's largest megacities (Mays et al., 2009; Wunch et al., 2009; Bergeron and Strachan, 2011; Levin et al., 2011; Silva et al., 2013; Hase et al., 2015; Vogel et al., 2019; Babenhauserheide et al., 2020). A lot of studies are based on the results of routine observations by the international ground-based monitoring networks: ICOS (ICOS, 2020), NOAA ESRL (NOAA ESRL, 2020), TCCON (TCCON, 2021), COCCON (COCCON, 2021), FLUXNET (FLUXNET, 2020). Also, national instrumental air quality control systems were used (Airparif, 2020) as well as the satellite measurement systems (Kuhlmann et al., 2019, Oda et al. 2018) and individual observational stations (Zinchenko et al., 2002; Pillai et al., 2011). It is important to mention

measurement campaigns organized in the framework of major scientific projects, such as InFLUX (sites.psu.edu/influx; Turnbull et al., 2014), Megacities Carbon Project (https://megacities.jpl.nasa.gov/portal/; Duren and Miller, 2012), MEGAPOLI (http://www.megapoli.info, Lopez et al., 2013), CO2-Megaparis project in Paris (https://co2-megaparis.lsce.ipsl.fr, Bréon et al., 2015), COCCON – Paris (http://www.chasing-greenhouse-gases.org/coccon-in-paris/), and VERIFY (https://verify.lsce.ipsl.fr/). The important goal is to improve existing techniques and to develop new algorithms for the space-borne detection of the $CO_2$ plumes originating from intensive compact sources such as large cities and big thermal power plants (TPP) (Kuhlmann et al., 2019; SMARTCARB project, https://www.empa.ch/web/s503/smartcarb). Bovensmann et al. (2010) and Pillai et al. (2016) proposed to create and launch new specialised satellite instruments for studying natural and anthropogenic sources and sinks of carbon dioxide with high spatial resolution. At the same time, the variety of modelling tools used to simulate the atmospheric $CO_2$ fields and assimilate the results of observations is also quite large: ranging from simple mass balance models (Hiller et al., 2014; Zimnoch et al., 2010, Makarova et al., 2018) to modern transport and photochemical models (Ahmadov et al., 2009; Göckede et al., 2010, Pillai et al., 2011, Pillai et al., 2012).

The present study is focused on the $CO_2$ emission by St. Petersburg, Russian Federation. The area of St. Petersburg urban agglomeration is about 1440 $km^2$, while the city centre, which is characterized by high construction density, occupies 650 $km^2$. The city has a population of ~5.4 million people (the official data for 2019, St. Petersburg Center for Information and Analytics, 2020); according to unofficial data the population is now more than 7 million (Shevlyagina, 2020). The population density is ~3800 people/$km^2$ on average. It can reach ~7300 people/$km^2$ on the territories with high construction density (Solodilov, 2005). The data on total emissions of anthropogenic air pollutants in St. Petersburg are provided in the annual reports of the municipal Environmental Committee (Serebritsky, 2018; Serebritsky, 2019). Published data are based on the emission sources inventory method ("bottom-up") where $CO_2$ fluxes for urban areas are calculated on the basis of information about the landscape and the type of anthropogenic activity (e.g., number and type of buildings, location of roads, traffic intensity, the presence and type of TPP, etc.) using appropriate emission factors (Gurney et al., 2002; Serebritsky, 2018). On average, the contribution of St. Petersburg to the total greenhouse gas emissions of the Russian Federation is about 1%. According to official inventory data for 2015, the integral $CO_2$ emission from the territory of St. Petersburg is about 30 Mt/year and the inter-annual variability of this estimate in the period 2011-2015 did not exceed 1 Mt/year (Serebritsky, 2018). In the mentioned official inventory report, it is noted that most of St. Petersburg's emissions (more than 90%) are associated with power production. These estimates differ, for example, from the results obtained in the study of the structure of anthropogenic $CO_2$ emissions by the city of Baltimore (Maryland, USA): Roest et al. (2020) have reported that electricity production in Baltimore emits only 9% of $CO_2$ and the main part of emissions is related to transport (automobile 34%, marine 4%, air and rail transport 2%), as well as to the commercial sector (20%), industry (19%) and private residential housing (12%).

The main anthropogenic source of $CO_2$ is associated with the consumption of fossil fuels. However, a number of studies have demonstrated that for the territories with high population density carbon dioxide produced by human respiration process can make a significant contribution to total emissions (Bréon et al., 2015; Ciais et al., 2007; Widory and Javoy, 2003). According to some estimates, one person emits by breathing on average 1 kg of $CO_2$ per day (Prairie and Duarte, 2007), which would amount to about 3 Mt of $CO_2$ per year for St. Petersburg. Bréon et al. (2015) have shown that for Paris

the $CO_2$ emission from human breathing constitutes 8% of the total inventory emissions of the metropolis due to the use of fossil fuels.

        Official inventory ("bottom-up") estimates of the $CO_2$ emissions for St. Petersburg (Serebritsky, 2018) may have significant uncertainties both in the estimates of integral emissions and in the data on the spatial and temporal distribution of the $CO_2$ fluxes. This suggestion is confirmed, in particular, by the significantly different values of the CO-to-$CO_2$ emission

ratio (ER) for St. Petersburg obtained by Makarova et al. (2021) from the field measurements ($ER_{CO/CO2} \approx 6$ ppbv/ppmv) and calculated using the official emission inventory data reported by Serebritsky (2018) ($ER_{CO/CO2} \approx 21$ ppbv/ppmv). The $ER_{CO/CO2}$ ratio is one of the most important characteristics of the source of air pollution, since its value can indicate the nature of the emission. For cities, $ER_{CO/CO2}$ mostly reflects the structure of FF consumption.

        In 2019, the mobile experiment EMME (Emission Monitoring Mobile Experiment) was carried out on the territory of

the St. Petersburg agglomeration with the aim to estimate the emission intensity of greenhouse ($CO_2$, $CH_4$) and reactive (CO, $NO_x$) gases for St. Petersburg (Makarova et al., 2021). St. Petersburg State University (Russia), Karlsruhe Institute of Technology (Germany) and the University of Bremen (Germany) jointly prepared and conducted this city campaign. The core instruments of the campaign were two portable FTIR (Fourier Transform InfraRed) spectrometers Bruker EM27/SUN which were used for ground-based remote sensing measurements of the total column amount of $CO_2$, $CH_4$ and CO at upwind

and downwind locations on opposite sides of the city. The applicability and efficiency of this measurement scenario and EM27/SUN spectrometers have been shown by Hase et al. (2015), Chen et al. (2016), Dietrich et al., (2021). The description of the EMME experiment has been given in full detail in the paper by Makarova et al. (2021). This study has also reported the estimations of the area fluxes for the emissions of $CO_2$, $CH_4$, $NO_x$ and CO by St. Petersburg. In 2020, the EMME experiment was continued. It started in March before the COVID-19 pandemic lockdown and consisted of six days of field

measurements (three days before the lockdown and three days during the lockdown).

        The present study continues the analysis of the data of EMME-2019 and demonstrates the first results of the 2020 campaign. We concentrate our efforts on the $CO_2$ emissions leaving the results relevant to other gases beyond the scope of the research. As an extension to the work by Makarova et al. (2021) our goal in this paper is to estimate the integral $CO_2$ emission by St. Petersburg megacity rather than area fluxes. Completing this task consists of the following basic steps:

-     We use mobile FTIR measurements to obtain $CO_2$ column enhancements ($\Delta CO_2$) related to urban anthropogenic emissions.

-       We adapt the ODIAC database (Oda and Maksyutov, 2011) to construct a priori information on the spatio-temporal distribution of anthropogenic $CO_2$ emissions on the territory of St. Petersburg.

-       We initialize the HYSPLIT dispersion model, HYbrid Single-Particle Lagrangian Integrated Trajectories (Draxler and Hess, 1998; Stein et al., 2015) with the ODIAC emissions to simulate $CO_2$ 3D field over the city of St. Petersburg.

-       We evaluate the performance of our HYSPLIT model setup by calculating the surface $CO_2$ concentrations and comparing them with the routine in-situ measurement results (Foka et al., 2019).

-       We scale the emission input data for the HYSPLIT model simulations in order to reproduce the observed $\Delta CO_2$.

-       Finally, from the scaled emission a priori data we get the estimate of integral $CO_2$ emission by St.Petersburg.

The paper is organized as follows. Section 2 describes the methods and instruments, including a description of the EMME measurement campaign and the equipment used, methods for processing the measurement results, the configuration of the HYSPLIT model and its evaluation based on calculations of ground-level $CO_2$ concentrations. Section 3 presents main results of EMME-2019 and EMME-2020 including estimates of integrated $CO_2$ emissions derived from FTIR measurements of the 2019 and 2020 field campaigns, combined with HYSPLIT model simulations. Section 4 contains a summary of our findings.

## 2 Methods and instrumentation

The main goal of the EMME measurement campaigns in 2019 and 2020 organized jointly by SPbU (St. Petersburg State University, Russia), KIT (Karlsruhe Institute of Technology, Germany) and UoB (University of Bremen, Germany) was to evaluate emissions of $CO_2$, $CH_4$, CO and $NO_x$ from the territory of St. Petersburg. Similar to 2019, the EMME-2020 campaign was conducted in spring (March - early May). This time of the year is preferable for a successful study of urban emissions, especially $CO_2$, due to the following reasons: (1) a daylight duration is sufficient for FTIR remote sensing measurements; (2) the influence of vegetation processes on the daily evolution of the $CO_2$ concentration in the atmosphere is negligible; (3) the winter heating of the city buildings is still active which is a significant source of the $CO_2$ emissions for northern cities such as St. Petersburg. In contrast to the 2019 campaign, when two mobile EM27/SUN FTIR spectrometers were used in the field experiment for simultaneous measurements inside and outside of the air pollution plume, all measurements in 2020 were performed with one spectrometer which was moved between clean and polluted locations within one day. In 2019, the field measurements were carried out during 11 days in total, and on 6 days in 2020. The number of observations in 2020 was smaller than in 2019 due to the quarantine restrictions related to the COVID-19 pandemic. These restrictions were imposed in St. Petersburg on 28 March, 2020. During several days of the 2020 campaign, measurements inside the city pollution plume were made at two locations, which allowed to increase the total number of observations. Details of both field campaigns are given in Tables 1 and 2 for 2019 and 2020, respectively. The tables contain the Fourier transform spectrometer (FTS) instrument IDs (#80 and #84 in 2019, #84 in 2020), the position on the upwind and downwind sides of the city (latitude and longitude), and the duration of observations. Note that each experiment presented in the tables

consists of a pair of series of measurements – from the upwind and downwind sides. In 2019, observations of two FTS instruments (#80 and #84) simultaneously were used for this purpose (see Table 1). In 2020 the single FTS instrument (#84) was moved between the upwind and downwind positions (see Table 2). The average duration of measurements in 2019 was 3 hours within the period of ~12:00-15:00. In 2020, the duration of the measurements was limited to about 1 hour (sometimes less), and the observation time varied from 11:00 to 19:00. Since a single instrument was used in 2020, the time

difference between upwind and downwind measurements in 2020 ranged from 3 to 5 hours.

## 2.1 Bruker EM27/SUN FTS and spectra processing

Bruker EM27/SUN (Gisi et al., 2012; Frey et al., 2015, Hase et al., 2016) is a portable robust FTS having low spectral resolution of 0.5 cm$^{-1}$. It was designed for accurate and precise ground-based observations of $CO_2$, $CH_4$ and CO column-

170 averaged abundances ($X_{CO2}$, $X_{CH4}$ and $X_{CO}$) in the atmosphere. These FTIR spectrometers were used to build the COCCON network (COCCON, 2021; Frey et al., 2019). EM27/SUN is equipped with a Camtracker, a solar tracking system developed by KIT (Gisi et al., 2011). A Camtracker consists of an altazimuthal solar tracker, a USB digital camera and "CamTrack" software which processes an image acquired by a camera and controls the tracker's movement. EM27/SUN FTS is designed on the basis of a robust RockSolid™ interferometer having high thermal and vibrational stability; the detailed description of

175 the instrument is given by Gisi et al. (2012). Therefore, this type of instruments is being successfully implemented for setting up fully automated stationary city network MUCCnet (Munich Urban Carbon Column network, Dietrich et al., 2021) and for performing a number of mobile campaigns (Klappenbach et al., 2015; Luther et al., 2019; Makarova et al., 2021).

In our study, we used the dual-channel EM27/SUN with quartz beamsplitter. Additionally, two detectors allow observing $X_{CO}$ and future improvements of the $X_{CO2}$ retrieval (Hase et al., 2016). FTS registers an interferogram which is the

180 Fourier transform of the infrared spectrum of direct solar radiation. The processing of data acquired by EM27/SUN spectrometer consists of the following stages:

- deriving spectra from raw interferograms including a DC-correction and quality assurance procedures (Keppel-Aleks et al., 2007);

- deriving $O_2$, $CO_2$, CO, $H_2O$, and $CH_4$ total columns (TCs) from FTIR spectra by scaling a priori profiles of retrieved gases

(Frey et al., 2019; COCCON, 2021).

To process the spectral data we used the free software PROFFAST which had been specially developed for COCCON network (COCCON, 2021; Frey et al., 2019). PROFFAST has been developed by KIT in the framework of several ESA projects for processing the raw data delivered by the EM27/SUN FTS. For the retrievals of total columns (TCs) of target species the following spectral bands are used (Frey et al., 2015; Hase et al., 2016; Frey et al., 2019): 4210-4320 cm$^{-1}$ (target

gas – CO, interfering gases – $H_2O$, HDO, $CH_4$), 5897-6145 cm$^{-1}$ (target gas – $CH_4$, interfering gases – $H_2O$, HDO, $CO_2$), 6173-6390 cm$^{-1}$ (target gas – $CO_2$, interfering gases – $H_2O$, HDO, $CH_4$), 7765-8005 cm$^{-1}$ (target gas – $O_2$, interfering gases –

$H_2O$, HF, $CO_2$), and 8353-8463 cm$^{-1}$ (target gas – $H_2O$). The retrieval algorithm requires the following input: temperature profile in the atmosphere, pressure at the ground level, and the a priori data on the mole fraction vertical distribution of the atmospheric trace gases. These data are generated by the TCCON network software which ensures their compatibility over the TCCON network (TCCON, 2021). The close agreement of EM27/SUN observations analyzed with PROFFAST with a collocated TCCON spectrometer has been demonstrated in the framework of the ESA project FRM4GHG (Sha, 2020).

In order to obtain a reliable value of the $CO_2$ emission for St. Petersburg, it is necessary to eliminate possible systematic error caused by the instrument bias. This goal was reached by carrying out a cross-calibration of the instruments. In April-May 2019 both instruments passed a four day cross-calibration. The comparison of side-by-side measurements of $X_{CO2}$ by FTS#80 and FTS#84 allowed determining calibration factor which was used for converting $X_{CO2}$ measured by FTS#80 to the scale of FTS#84. Detailed information about side-by-side calibration of FTIR-spectrometers is given in the paper by Makarova et al. (2021).

## 2.2 A priori data on FF $CO_2$ emissions

The global emission inventory ODIAC (Oda and Maksyutov, 2011; Oda, Maksyutov and Andres, 2018) is used in the present study for characterisation of the area fluxes of the $CO_2$ emission from the territory of St. Petersburg and its suburbs. ODIAC provides global information on monthly average $CO_2$ emissions due to consumption of fossil fuels. The high spatial resolution of ODIAC (1 km × 1 km) is achieved through a joint interpretation of the existing global inventory of anthropogenic $CO_2$ sources, data on FF consumption, and satellite observations of the night-time glow of densely populated areas of the Earth. We use the data for 2018 emissions given in the ODIAC2019 version (Oda and Maksyutov, 2020).

The $CO_2$ emission data have been extracted from the ODIAC database for the domain that includes St. Petersburg and its suburbs (59.60-60.29° N, 29.05-31.33° E, Fig. 1). The sources of anthropogenic $CO_2$ emissions are concentrated within the administrative borders of the city. Most of these sources have intensities of ~4000 tons/month/km$^2$ and higher and are located within the borders of the city ring road. Summing up the ODIAC data within the city borders gives an estimate of the average integrated $CO_2$ emission of ~2710 kt per month with variations from 2429 kt in July to 3119 kt in March (Fig. 2). The emissions are maximal in late winter and early spring, and are minimal in summer. In general, the seasonal variability of emissions is insignificant (~8%), therefore the data for 12 months of 2018 were averaged in order to obtain an estimate of the mean annual distribution of urban $CO_2$ emissions. The integrated annual emission of St. Petersburg equals to 32529 kt, which is in good agreement with published official estimates: about 30 million tons for the period from 2011 to 2015 (Serebritsky, 2018).

The nominal latitude/longitude size of the ODIAC data pixel is 30 arcseconds (Oda and Maksyutov, 2011), which defines a global spatial resolution of about 1 km × 1 km. Since the length of a degree of longitude changes with the latitude,

the pixel size for St. Petersburg (~60° N) is smaller and equals to 0.93 km × 0.46 km (0.43 km$^2$). It should be noted that the average annual urban emission flux is ~26 kt km$^{-2}$ while in the central part of the city it can reach up to 80 kt km$^{-2}$. There is one pixel in the ODIAC data located in the centre of St. Petersburg with an extremely high emission flux of 7000 kt km$^{-2}$.

Normally, power plants and industrial enterprises manifest themselves as point sources of strong emission. However, we cannot confidently attribute this particular ODIAC pixel to any source of this type, since there is no such object near it. There are about a dozen of large thermal power plants on the territory of St. Petersburg, but all of them appear to be rather far from this location. Despite the lack of published data on anthropogenic $CO_2$ emissions at the city scale, we were able to explore detailed reports from municipal inventories of stationary air pollution sources (unpublished, but available on

request). According to the inventories of $NO_x$, CO, $SO_2$, $NH_3$, VOC and PM10 pollutants, there are no stationary objects of an extreme emission close to the point of our interest. Thus we feel confident to smooth out this outlier and replace it by the value averaged over the neighboring ODIAC pixels. As a result, it amounted to 42 kt km$^{-2}$.

## 2.3 The HYSPLIT model general setup

The spatial and temporal evolution of the urban pollution plume was simulated using the HYSPLIT model (Draxler and

235 Hess, 1998; Stein et al., 2015). Calculations were performed for the territory of the St. Petersburg agglomeration using the offline version of the HYSPLIT model with the setup similar to the one that was successfully used previously for the $NO_x$ plume modelling (Ionov and Poberovskii, 2019; Makarova et al., 2021). A 3-dimensional field of anthropogenic air pollution was calculated for a spatial domain with coordinates 54.8°-61.6° N, 23.7°-37.8° E; the domain grid size was 0.05°×0.05° latitude and longitude (see Fig. 3, top). The vertical grid of the model is set to 10 layers with the altitude of the

240 upper level at 1, 25, 50, 100, 150, 250, 350, 500, 1000 and 1500 meters a.s.l., respectively. As a source of meteorological information (vertical profiles of the horizontal and vertical wind components, temperature and pressure profiles, etc.), the NCEP GDAS (National Centers for Environmental Prediction Global Forecast System) data were used, presented on a global spatial grid of 0.5° × 0.5° latitude and longitude with time interval of 3 hours (NCEP GDAS, 2020).

To run HYSPLIT we used the software package: HYSPLIT 4, June 2015 release, subversion 761. The advanced setup

of the HYSPLIT model was configured as follows (basic parameters):

- default method of vertical turbulence computation,

- horizontal mixing computed proportional to vertical mixing,

- boundary layer stability computed from turbulent fluxes (heat and momentum),

- vertical mixing profile set variable with height in the planetary boundary layer (PBL),

- boundary layer depth set from the meteorological model (input meteorology data),

- puff mode dispersion computation with a "tophat" concentration distribution on a horizontal and vertical scale.

Spatial distribution of FF $CO_2$ emission sources and their intensities are taken from the ODIAC database. The original ODIAC data were converted into a set of larger pixels (~3.6 km$^2$). Pixels with the area fluxes lower than 8 kt km$^{-2}$ have been filtered out in order to keep only the urban sources which could be attributed to the St. Petersburg agglomeration. The resulting array which was used as the input for HYSPLIT consisted of 376 pixels and is shown in Fig. 3 (bottom). The integral $CO_2$ emission that corresponds to this array equals to 26316 kt year$^{-1}$; this is the value being used as a HYSPLIT first guess hereafter.

## 2.4 Test simulations of ground-level $CO_2$ concentrations

Routine measurements of $CO_2$ surface concentrations have been carried out at the atmospheric monitoring station of St. Petersburg University in Peterhof (59.88° N, 29.82° E) since 2013. These observations are the in situ measurements using a gas analyzer Los Gatos Research GGA 24r-EP. The instrument is installed on the outskirts of a small town of Peterhof in the suburbs of St. Petersburg (see location in Fig. 1). This place is far enough away from busy streets and other local sources of pollution, with an ambient air intake being 3 meters above the surface. To test the HYSPLIT model setup for the St. Petersburg region, we simulated the surface concentration of $CO_2$ near Peterhof during the 2019 EMME measurement campaign – from March 20 to April 30, 2019. The results of the model calculations were compared to the data of in situ measurements (due to the instrument failure in 2020 the comparison is limited to the period of EMME campaign in 2019 only). Observational data and simulation results were averaged over 3-hour intervals. The resulting comparison is shown in Fig. 4. The model reproduces the temporal variations of $CO_2$ including the main periods of significant growth of concentration; the correlation coefficient between the calculation and measurements is equal to 0.72. The background value of the surface concentration is taken as 415 ppmv based on the local measurements (415 ± 2 ppmv is the mean value of diurnal minima during the campaign from March 20 to April 30, 2019). It is important to emphasize that quantitative agreement is achieved by linear scaling of the a priori integral urban $CO_2$ emission. The scaling coefficient for emissions corresponds to the value of the integral urban $CO_2$ emission from the territory of St. Petersburg of 44800±1900 kt year$^{-1}$ (the given uncertainty is due to the uncertainty of the fitted scaling factor). This value is noticeably higher than official estimates mentioned above and ODIAC data for 2018 (32529 kt). The average discrepancy between the measurement and simulation data shown in Fig. 4 is 2±9 ppmv (model calculations are systematically lower).

## 3 Evaluation of integrated $CO_2$ emissions from field FTIR measurements
## 3.1 The results of the EMME-2019 campaign

We simulated the $CO_2$ total column (TC) spatial distributions over the territory of the St.Petersburg agglomeration for the time periods of FTIR mobile measurements conducted in the framework of the EMME-2019 experiment in March-April 2019. Obviously, the anthropogenic contribution to the $CO_2$ TC is concentrated mostly in the lower boundary layer, with a

top height of ~200 to ~1600 m. Therefore, HYSPLIT model was configured to simulate $CO_2$ concentrations at 10 altitude levels (0-1500 m), which were then integrated to obtain the $CO_2$ column in the boundary layer. The maps of the $CO_2$ plume obtained in this way show that for all the analyzed experiments, the choice of the location of the upwind and downwind measurement points was correct (see Appendix A, Fig. A1). The differences between the results of FTIR measurements of the $CO_2$ TC inside and outside the pollution plume ($\Delta CO_2$) were compared with the differences in the $CO_2$ column in the boundary layer simulated by HYSPLIT at the corresponding locations. HYSPLIT calculations were performed with a temporal resolution of 15 minutes. The data series of measured and calculated $CO_2$ contents for the experiments involved in the analysis are shown in separate plots in the Appendix B, Fig. B1. It is clearly seen from the plots that the "downwind-upwind" enhancements in $CO_2$ observed by the measurements are significantly higher than predicted by HYSPLIT, which indicates an underestimation of inventory $CO_2$ emissions. An example of simulated $CO_2$ plume and a time series of $CO_2$ total column measurements and HYSPLIT calculations for a typical day of experiments in 2019 (April 4) is given in Fig. 5. For the sake of comparison, the simulation results and measurement data were averaged over time periods of field observations (the duration of each experiment is given in Table 1).

In order to obtain a quantitative agreement between simulated and observed $\Delta CO_2$, the inventory data (the ODIAC emissions), which are used as input information for the HYSPLIT dispersion model, should be scaled (Flesch et al., 2004). The scaling factor (*SF*) is derived as follows. The data from all days of measurements are compared to the corresponding model simulations, see Fig. 6 as an example of a scatter plot (left panel). The scaling factor is determined as a slope value of the following regression line (e.g. the slope is $2.88 \pm 0.21$, as shown in Fig. 6):

$$\Delta CO_2[FTIR]_i = SF \times \Delta CO_2[HYSPLIT^{ODIAC}]_i \tag{1}$$

where $\Delta CO_2[FTIR]_i$ is is the difference between the downwind and upwind FTIR measurements averaged over the duration of experiment *i* (see Table 1 and Table 2, Appendix A and Appendix B for the details of every field experiment) and $\Delta CO_2[HYSPLIT_{ODIAC}]_i$ is the averaged difference between the downwind and upwind $CO_2$ column calculated using the HYSPLIT dispersion model for the location and time of FTIR observations, and initialized with ODIAC $CO_2$ emissions.

The error assessment for the scaling factor should be discussed in some detail. The $1\sigma$ precision for the $XCO_2$ individual measurement is of the order of 0.01 %–0.02 % (<0.08 ppm) (e.g. Gisi et al., 2012; Chen et al., 2016; Hedelius et al., 2016; Klappenbach et al., 2015; Vogel et al., 2019). The error of the scaling factor was estimated under the assumption that the measurement errors are the same for all days as well as the model simulation errors. The error bars indicated in Fig. 6 as boxes are in fact the variations of $\Delta CO_2$ obtained as standard deviation of observations and simulations within one observational series (see Appendix B, Fig. B1). Obviously, these quantities comprise both measurement errors and simulation errors (including those associated with wind direction and speed uncertainty), and temporal variability of the $CO_2$ TC. One can see that these quantities differ from day to day.

The right panel of Fig. 6 demonstrates that the model reproduces well the evolution of $\Delta CO_2$ recorded in field measurements; the correlation coefficient between the results of modelling and observations is 0.94. The derived scaling factor yields the integral anthropogenic $CO_2$ emission value of $75800\pm5400$ kt year$^{-1}$; i.g. the value of 75800 results from the multiplication $26316\times2.88$ (the 2.88 here is the slope on the scatter plot, and 26316 is the model first guess, see Section 2.3). Resulting $CO_2$ emission rate is almost twice as high as the above estimate, based on the analysis of ground-level $CO_2$ measurement data (Section 2.4, $44800\pm1900$ kt year$^{-1}$). This difference may indicate a significant contribution of elevated $CO_2$ sources (industrial chimneys) that could not be registered by the ground-level in situ measurements, as the elevated exhausts of pollution are more likely to further rise up, rather than descend to the ground. In contrast, FTIR measurements of the total column keep being sensitive to this kind of emissions. In addition, while FTIR measurements implement a "cross section" of the urban pollution emission zone in a series of multidirectional trajectories (depending on the wind direction), local ground-level in situ measurements at a specific location (Peterhof) can not capture the contribution of the entire mass of urban emissions. Thus, estimates of integral $CO_2$ emissions based on the interpretation of ground-level measurements in Peterhof can be considered as a lower limit of an estimate.

An earlier analysis of the results of the EMME-2019 measurement campaign focused in particular on inferring the area fluxes of urban $CO_2$ emissions from St. Petersburg. In order to achieve this goal, the simplified mass balance approach was applied to the observed $CO_2$ enhancements ($\Delta CO_2$) which were attributed to the accumulation of pollution during the air mass movement on its way from the upwind side to the downwind side of the megacity (see Makarova et al., 2021 for full details). Basically, the mass balance approach was adopted in the form of a one-box model, and the area flux $F$ was calculated using the following equation:

$$F = \frac{\Delta CO_2 \cdot V}{L}$$
(2)

where $F$ is the $CO_2$ area flux, $\Delta CO_2$ is the difference between the downwind and upwind FTIR measurements, $V$ is the mean wind speed and $L$ is the mean path length of an air parcel which goes through the urban area (Chen et al., 2016). The obtained mean value of the $CO_2$ area flux was equal to $89\pm28$ kt yr$^{-1}$ km$^{-2}$ and was attributed to the emission from the city centre. As shown above, in the current study, the application of the HYSPLIT model allowed us to estimate the integral anthropogenic $CO_2$ emission of the entire megacity. In order to check the consistency with previous results, in the present study we made calculations of area fluxes on the air trajectories of field measurements using the ODIAC emission database scaled to the integral $CO_2$ emission derived from the results of EMME-2019 combined with the HYSPLIT simulations ($75800\pm5400$ kt year$^{-1}$). Schematically, the air trajectories corresponding to the 2019 FTIR measurement locations are shown in Fig. 7. These trajectories were simulated as backward trajectories by the HYSPLIT model in the boundary layer of the atmosphere. The resulting values of anthropogenic $CO_2$ area fluxes calculated by integrating the ODIAC data along these trajectories, are shown in Fig. 8 in comparison with the experimental estimates by Makarova et al., 2021. As in the study by

Makarova et al., 2021, the width of the air paths was assumed to be 10 km. On average, according to ODIAC data, the area flux for the 2019 measurement trajectories was $106\pm9$ kt yr$^{-1}$ km$^{-2}$, that is somewhat higher than the experimental estimates ($89\pm28$ kt yr$^{-1}$ km$^{-2}$) but agree within the error limits. Significantly higher variability in the experimental data may be related to the variability of the wind field, which is not taken into account in the simplified mass balance approach.

## 3.2 The results of EMME-2020 and comparison with EMME-2019

The data of mobile FTIR measurements performed in March-early May 2020 were processed and analysed in the same way as it was done for data acquired during the measurement campaign in 2019. An example of simulated $CO_2$ plume and a time series of $CO_2$ total column measurements and HYSPLIT calculations for a typical day of experiments in 2020 (April 8) is given in Fig. 9. The comparison of the observed and simulated mean values of $\Delta CO_2$ is shown in Fig. 10. Similar to the results of 2019, the HYSPLIT simulations reproduce well the observed evolution of $\Delta CO_2$. The correlation coefficient between the simulations and observations is 0.78. The estimation of the $CO_2$ emission was done using the described above approach based on the pollution plume modelling by HYSPLIT and scaling the ODIAC data which were taken as an a priori guess. For the EMME-2020 campaign, the derived integral anthropogenic $CO_2$ emission is $68400\pm7100$ kt yr$^{-1}$, which is about 10% lower than the estimate obtained for 2019 ($75800\pm5400$ kt yr$^{-1}$).

It should be noted that one can expect lower anthropogenic $CO_2$ emissions in the 2020 measurement data compared to the same period in 2019, since restrictive measures were imposed in St. Petersburg on March 28 due to the COVID-2019 pandemic. A number of studies have already reported significant reductions of air pollution that followed the lockdown events in different regions of the world (see e.g. Petetin et al., 2020; Pathakoti et al., 2020; Koukouli et al., 2020). According to Yandex data (https://yandex.ru/covid19/stat) the traffic intensity in the city of St. Petersburg decreased to 12-26% of the usual value on weekdays in the first week of quarantine (from March 30 to April 3) and amounted to 28-33% in the following week (from April 6 to April 10). Since we have no official data on the $CO_2$ emissions by traffic at our disposal, we used the average estimate for European countries, according to which the contribution of traffic to total emission constitutes 30% (European Parliament News, 2020). Under this assumption, a reduction in traffic activity down to 30% of the normal level should result in a reduction in total anthropogenic $CO_2$ emissions by 21% ($(1.0-(0.7+0.3\times0.3))\times100\%$).

The weak response of urban $CO_2$ emissions to restrictive quarantine measures may indicate a relatively small contribution of traffic to the total $CO_2$ emissions from the territory of St. Petersburg. This may be due to the higher contribution of emissions associated with residential heating (including consumption of natural gas in private residences, e.g. stoves and water boilers), which is more important for such a northern city as St. Petersburg, unlike many European cities. Normally, the heating is still working in St. Petersburg in March and April, and the corresponding $CO_2$ emissions cannot be reduced due to the quarantine. Our confident expectation to detect the transport contribution is based on the high sensitivity of FTIR measurements of $X_{CO_2}$ using EM27/SUN spectrometers and COCCON methodology. If the emission from traffic

were higher it would have been definitely identified during the campaign. The high sensitivity of our measurements to the $CO_2$ pollution from different sources is demonstrated by the following examples. The results of EMME-2019 revealed that the emission of a single TPP located on the north-eastern side of the city (see Fig. 11) can add $\sim 5 \times 10^{19}$ molecules/cm$^2$ to the $CO_2$ TC (Makarova et al., 2021). During the 2020 measurement campaign, one of the series of FTIR measurements was performed near the Waste Processing Plant (WPP) on the eastern side of the city (see Fig. 11). The contribution of this local

$CO_2$ source was $\sim 1 \times 10^{19}$ molecules/cm$^2$. We emphasise that these measurements, being significantly affected by local sources, were excluded from statistical analysis. In general, for these reasons (including unfavorable weather conditions and wrong location of FTIR measurement points), data from only a few experiments were excluded: No.8 on April 18, 2019, No.10 on April 25, 2019, No.11 on April 30, 2019 (see Table 1) and No.4 on March 27, 2020 (see Table 2). However, the given examples indicate the crucial role of stationary, non-transport sources of emissions, which were not subject to

restrictive quarantine measures.

A thorough analysis of all experiments performed during the 2019 and 2020 measurement campaigns has shown that there were days with similar air trajectories and similar downwind measurement locations. These situations occurred twice: on March 27, 2019 and April 5, 2020, and on April 1, 2019 and April 8, 2020 (see Fig. 11). Both series of 2020 measurements, on April 5 and April 8, were performed during the COVID-19 quarantine period. We calculated the $CO_2$ area

fluxes for these days applying the mass balance approach which was used by Makarova et al., 2021. The results are presented in Table 3. Unexpectedly, the estimates indicate an increase of area fluxes during the quarantine period in 2020, compared to the same period in 2019. According to the data of weather archive (http://rp5.ru/Weather_archive_in_Saint_Petersburg, last access 3 November 2020), the mean ambient temperature in St. Petersburg was +5.0 °C and +3.2 °C for the period from March 27 to April 8 in 2019 and 2020, accordingly. Thus,

somewhat colder weather in 2020 may contribute to the increase of $CO_2$ emission due to the more intense residential heating. However, the high uncertainty of the $CO_2$ area flux estimates due to the uncertainties of the wind field and of the effective path length (for details, see Makarova et al., 2021) does not allow us to gain sufficient confidence in the nature of the detected differences.

To our opinion, the most important finding of our study based on the analysis of two observational campaigns is a

400 significantly higher $CO_2$ emission from the megacity of St. Petersburg as compared to the data of municipal inventory: $\sim 75800 \pm 5400$ kt yr$^{-1}$ for 2019, $\sim 68400 \pm 7100$ kt yr$^{-1}$ for 2020 versus $\sim 30000$ kt yr$^{-1}$ reported by official inventory. Besides, this finding is consistent with the estimate of the $CO_2$ emission area flux by Makarova et al., 2021 which was about double of the EDGAR inventory for St. Petersburg (EDGAR, 2019). The difference can be partly explained by the impact of diurnal and seasonal variations of anthropogenic activity, since our measurements were conducted during the period of maximum

$CO_2$ emission (early spring and afternoon) and therefore represent the upper limit of the emission estimates. According to the ODIAC data (see Fig. 2) emissions in March and April have to be scaled down by the factor of $\sim 1.07$ to represent the annual

average. The global database of hourly scaling factors (Nassar et al. 2013) gives also a factor of ~1.07 for St. Petersburg to scale down the afternoon emission rates to the daily average. So, dividing our estimates twice by 1.07 gives ~59000÷66000 kt yr$^{-1}$, which is still higher than the official inventory value. Compared to other world cities, the integral $CO_2$ emission of St. Petersburg is not that high – e.g, the ODIAC inventory reports: ~18000 kt yr$^{-1}$ for San Francisco, ~37000 kt yr$^{-1}$ for Paris, ~51000 kt yr$^{-1}$ for Mexico, ~88000 kt yr$^{-1}$ for Delhi, ~106000 kt yr$^{-1}$ for Moscow, ~136000 kt yr$^{-1}$ for Hong Kong, ~172000 kt yr$^{-1}$ for Tokyo and ~227000 kt yr$^{-1}$ for Shanghai (the data are taken from the paper by Umezawa et al., 2020, Fig. 3). Typically, these estimates of urban $CO_2$ emissions are strongly correlated with the city's population – e.g. ~1 million people at San Francisco and ~23 million people at Shanghai.

**4 Summary and conclusions**

In 2019 and 2020, in spring, the mobile experiment EMME (Emission Monitoring Mobile Experiment) was carried out on the territory of St. Petersburg, which is the second largest industrial city in Russia with a population of more than 5 million . In 2020, several measurement series were obtained during the lockdown period caused by the COVID-19 pandemic. Previously, the $CO_2$ area flux has been obtained from the data of the EMME-2019 experiment using the mass balance approach. The present study is focused on the derivation of the integral $CO_2$ emission from St. Petersburg by combining the results of the EMME observational campaigns of 2019 and 2020 and the HYSPLIT model simulations. The ODIAC database is used as the source of the a priori information on the $CO_2$ emissions for the territory of St. Petersburg.

A number of studies (Pillai et al., 2016; Broquet et al. 2018; Kuhlmann et al., 2019; Babenhauserheide et al., 2020) have shown that emissions from large $CO_2$ sources (cities, thermal power plants) can be characterized by the difference between the results of measurements of the carbon dioxide concentration in the dry  atmospheric  column inside and outside of the pollution plume ($\Delta XCO_2$). The results of measurement campaigns in 2019 and 2020 have shown that for St. Petersburg in a set of mobile experiments the values of $\Delta XCO_2$ averaged over the duration of FTIR observations were in the range of 0.05-4.46 ppmv. For comparison, similar studies revealed the following values of $\Delta XCO_2$: 0.16-1.03 ppmv for Berlin, Germany (Kuhlmann et al., 2019), 0.80-1.35 ppmv for Paris, France (Pillai et al., 2016; Broquet et al. 2018), and 0-2 ppmv for Tokyo, Japan (Babenhauserheide et al., 2020). So, for St. Petersburg, the highest values of $\Delta XCO_2$ were detected (4.46 ppmv), if compared to similar measurements in Berlin, Paris and Tokyo. It should be noted that the value of $\Delta XCO_2$ depends not only on the integral emission of the source, but also on its spatial allocation (compact or distributed), the geometry of the field experiment (location of observations relative to the pollution plume) and on the meteorological situation during the measurements. This is why dispersion modeling, taking into account inventories of emission sources, is the most appropriate tool for interpreting the results of such observations.

The HYSPLIT model coupled with the scaled input from the ODIAC database reproduces well the results of FTIR observations of the $CO_2$ TC during both campaigns: the correlation coefficient between the results of modelling and

observations is 0.94 for 2019 and 0.78 for 2020. Lower value of the correlation coefficient for 2020 can be partly explained by the change in the spatial distribution of the $CO_2$ emission sources during the COVID-19 pandemic lockdown which could differ from the ODIAC distribution of the FF $CO_2$ sources. However, the number of data is not sufficient to confirm this suggestion. The most important finding of the study based on the analysis of two observational campaigns is a significantly higher $CO_2$ emission from the megacity of St. Petersburg as compared to the data of municipal inventory: ~75800±5400 kt yr$^{-1}$ for 2019, ~68400±7100 kt yr$^{-1}$ for 2020 versus ~30000 kt yr$^{-1}$ reported by official inventory. The comparison of $CO_2$ emissions obtained during the COVID-19 lockdown period in 2020 to the results obtained during the same period of 2019 demonstrated a decrease in emission of 10% or 7400 kt yr$^{-1}$.

There was an attempt to simulate the in situ measurements of the $CO_2$ concentration performed at the observational site located in the suburb of the St. Petersburg megacity during the two-month period (March-April 2019). In this case the correlation coefficient between model simulations and observations was 0.72. In contrast to the estimates of the $CO_2$ emissions from FTIR measurements presented above, the simulation of in situ measurements gives a much lower value (by a factor of 1.5-1.7) of the $CO_2$ integrated emission: 44800±1900 kt year$^{-1}$. Similar differences were previously found between estimates of the $CO_2$ area fluxes for the central part of St. Petersburg, obtained both from the analysis of FTIR measurements, and from in situ measurements of $CO_2$ concentration (Makarova et al., 2021). This fact may indicate a significant contribution of elevated $CO_2$ sources (industrial chimneys) that could not be registered by the ground-level in situ measurements (in contrast to FTIR measurements of the total column). The approach of monitoring the outflows of large cities using arrays of compact FTIR spectrometers seems a promising and cost-effective route for assessing and monitoring the $CO_2$ emissions of these important sources. Recurring campaigns performed over extended periods or even the erection of permanent observatories as demonstrated by Chen et al. (Dietrichet et al., 2021) should be recognized as crucial components of strategies aiming at improved observational capacity for greenhouse gases on regional and urban domains.

**Data availability**

The datasets containing the EM27/SUN measurements during EMME-2019 and EMME-2020 can be provided upon request; please contact Maria Makarova (m.makarova@spbu.ru) and Frank Hase (Frank.Hase@kit.edu)

**Author contributions**

DVI and MVM conceived the study. MVM, DVI, FH, CA, VSK, SCF contributed greatly to the experimental part of the study. SCF, CA, and MVM were in charge of processing FTIR spectrometer data. DVI was in charge of numerical modelling by HYSPLIT. Together DVI, MVM, FH, TB, SCF, CA, VSK, and TW analysed and interpreted the results. DVI, MVM, and VSK prepared the original draft of the manuscript. Together DVI, MVM, FH, TB, SCF, CA, VSK, and TW reviewed and edited the manuscript.

**Competing interests**

The authors declare that they have no conflict of interest.

**Acknowledgements**

Two portable FTIR spectrometers EM27/SUN were provided to St. Petersburg State University, Russia, by the owner -
Karlsruhe Institute of Technology, Germany, in compliance with the conditions of temporary importation in the frame of the
VERIFY project. The procedure of temporary importation of the instruments to Russian Federation was conducted by the
University of Bremen, Germany. Ancillary experimental data were acquired using the scientific equipment of "Geomodel"
research centre of St. Petersburg State University. The authors acknowledge the participation of Anatoly V. Poberovskii in
the field measurement campaigns. The authors gratefully acknowledge the NOAA Air Resources Laboratory (ARL) for the
provision of the HYSPLIT transport and dispersion model used in this publication.

**Funding**

This activity has received funding from the European Union's Horizon 2020 research and innovation programme under grant
agreement No 776810 (VERIFY project).  This work was supported by funding from the Helmholtz Association in the
framework of MOSES (Modular Observation Solutions for Earth Systems). The development of the COCCON data
processing tools were supported by ESA in the framework of the projects COCCON-PROCEEDS and COCCON-
PROCEEDS II.  The research was supported by Russian Foundation for Basic Research through the project No.18-05-00011

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

**Table 1.** EMME-2019 field campaign details: the dates of experiments in 2019 and the locations of FTS instruments during the upwind and downwind observations. The data on the direction and speed of the surface wind correspond to observations at one of the meteorological stations in the center of St. Petersburg at local noon (http://rp5.ru/Weather_archive_in_Saint_Petersburg, last access: 11 March 2021).

| No. | Date | Wind speed, ms$^{-1}$ | Wind direction | FTS identifier (instrument #) location (latitude, longitude) observation time (local) | |
| --- | --- | --- | --- | --- | --- |
| | | | | upwind | downwind |
| 1. | 21 March 2019 | 3 | WSW | #80 59.88°N, 29.83°E 14:07-15:07 | #84 59.95°N, 30.59°E 13:08-15:36 |
| 2. | 27 March 2019 | 2 | WSW | #84 60.01°N, 29.69°E 11:49-15:08 | #80 59.85°N, 30.54°E 11:42-14:57 |
| 3. | 01 April 2019 | 3 | WSW | #84 60.01°N, 29.69°E 11:01-13:24 | #80 59.85°N, 30.54°E 11:15-14:31 |
| 4. | 03 April 2019 | 3 | S | #84 59.88°N, 29.83°E 14:47-16:02 | #80 60.04°N, 30.47°E 11:57-14:21 |
| 5. | 04 April 2019 | 3 | SW | #84 59.81°N, 30.09°E 11:59-14:16 | #80 60.04°N, 30.47°E 11:59-14:16 |
| 6. | 06 April 2019 | 2 | SE | no.84 59.95°N, 30.59°E 12:14-15:23 | no.80 60.01°N, 29.69°E 12:15-15:29 |
| 7. | 16 April 2019 | 2 | NE | #84 60.01°N, 29.69°E 11:13-15:08 | #80 59.86°N, 30.11°E 11:21-14:59 |
| 8. | 18 April 2019 | 2 | NE | #80 60.04°N, 30.47°E 12:07-14:56 | #84 59.81°N, 30.09°E 11:38-15:24 |
| 9. | 24 April 2019 | 1 | WSW | #84 60.01°N, 29.69°E 11:38-14:55 | #80 59.85°N, 30.54°E 11:52-15:22 |
| 10. | 25 April 2019 | 1 | WSW | #80 60.04°N, 30.47°E 12:07-14:49 | #84 59.81°N, 30.09°E 11:19-15:08 |
| 11. | 30 April 2019 | 2 | SSE | #80 59.85°N, 30.54°E 12:35-13:31 | #84 60.01°N, 29.69°E 12:22-13:46 |

**Table 2.** EMME-2020 field campaign details: the dates of experiments in 2020 and the locations of FTS instrument during the upwind and downwind observations. The data on the direction and speed of the surface wind correspond to observations at one of the meteorological stations in the center of St. Petersburg at local noon (http://rp5.ru/Weather_archive_in_Saint_Petersburg, last access: 11 March 2021).

| No. | Date | Wind speed, ms$^{-1}$ | Wind direction | FTS identifier (instrument #) location (latitude, longitude) observation time (local) | |
|---|---|---|---|---|---|
| | | | | upwind | downwind |
| 1. | 22 March 2020 | 1 | N | #84 60.11°N, 30.48°E 10:38-11:55 | #84 59.94°N, 30.40°E 13:17-14:38 |
| 2. | 22 March 2020 | 1 | N | #84 60.11°N, 30.48°E 10:38-11:55 | #84 59.81°N, 30.14°E 15:55-17:16 |
| 3. | 23 March 2020 | 2 | W | #84 59.93°N, 29.64°E 12:55-14:33 | #84 59.90°N, 30.52°E 16:24-18:02 |
| 4. | 27 March 2020 | 2 | WSW | #84 59.88°N, 29.83°E 10:35-11:51 | #84 59.94°N, 30.60°E 13:24-14:12 |
| 5. | 27 March 2020 | 2 | WSW | #84 59.88°N, 29.83°E 10:35-11:51 | #84 59.96°N, 30.60°E 14:34-15:15 |
| 6. | 05 April 2020 | 4 | WSW | #84 59.82°N, 29.96°E 12:44-13:43 | #84 59.83°N, 30.52°E 10:53-11:48 |
| 7. | 08 April 2020 | 3 | WSW | #84 59.89°N, 29.89°E 14:58-16:46 | #84 59.83°N, 30.52°E 11:09-13:43 |
| 8. | 01 May 2020 | 1 | ESE | #84 59.73°N, 30.25°E 18:01-19:03 | #84 60.05°N, 30.06°E 13:22-14:27 |
| 9. | 01 May 2020 | 1 | ESE | #84 59.73°N, 30.25°E 18:01-19:03 | #84 60.03°N, 30.00°E 15:10-16:11 |

**Table 3.** The $CO_2$ area fluxes (kt yr$^{-1}$ km$^{-2}$) obtained from mobile FTIR measurements in 2019 and 2020 which were performed under similar observational configurations.

| Measurement date | $CO_2$ area flux [kt yr$^{-1}$ km$^{-2}$] |
|---|---|
| 27/03/2019 | 76±60 |
| 05/04/2020 | 116±92 |
| 01/04/2019 | 48±38 |
| 08/04/2020 | 89±70 |

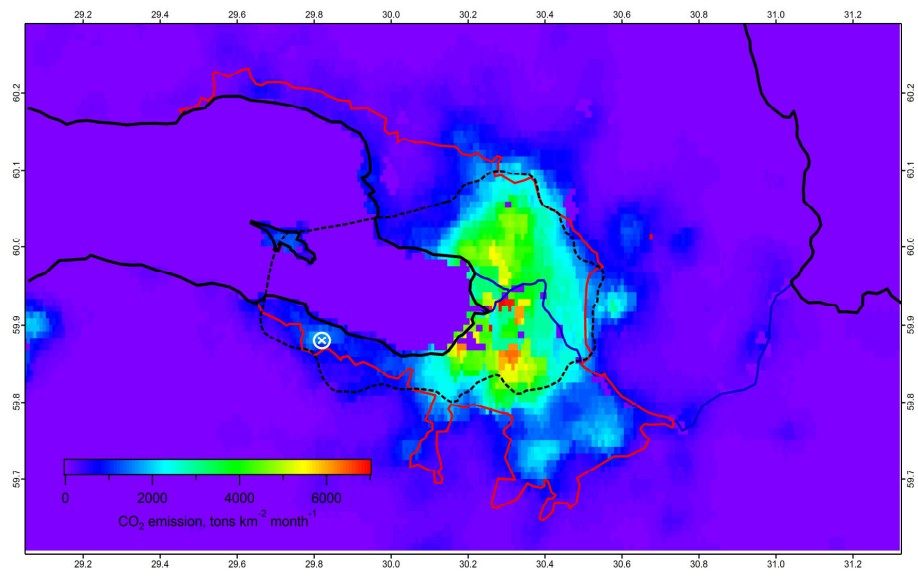

**Figure 1: Spatial distribution of anthropogenic CO$_2$ emission intensity on the territory of the St. Petersburg agglomeration (59.60-60.29° N, 29.05-31.33° E) according to ODIAC2019 data for April 2018. The red line indicates the administrative border of the city; the black dotted line indicates the city ring road. A white circle depicts the location atmospheric monitoring station of St. Petersburg University in Peterhof (see the text).**

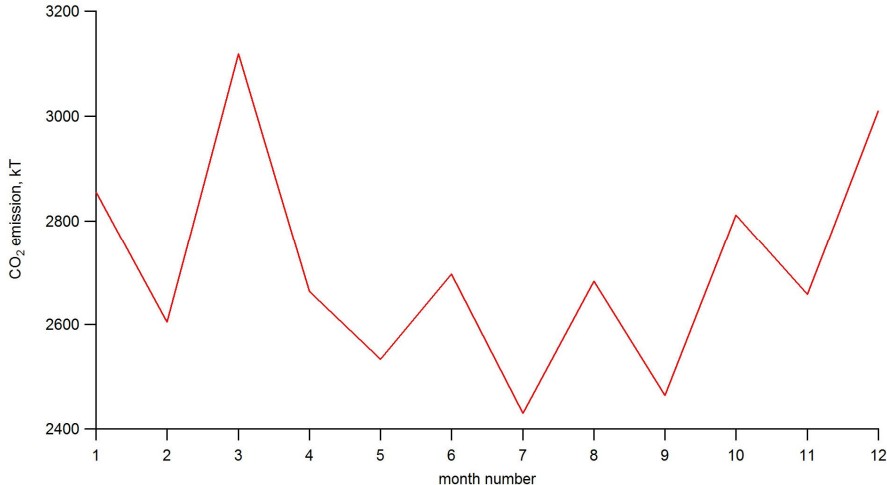

**Figure 2: Integrated monthly mean FF CO$_2$ emission from the territory of St. Petersburg according to ODIAC2019 data in 2018.**

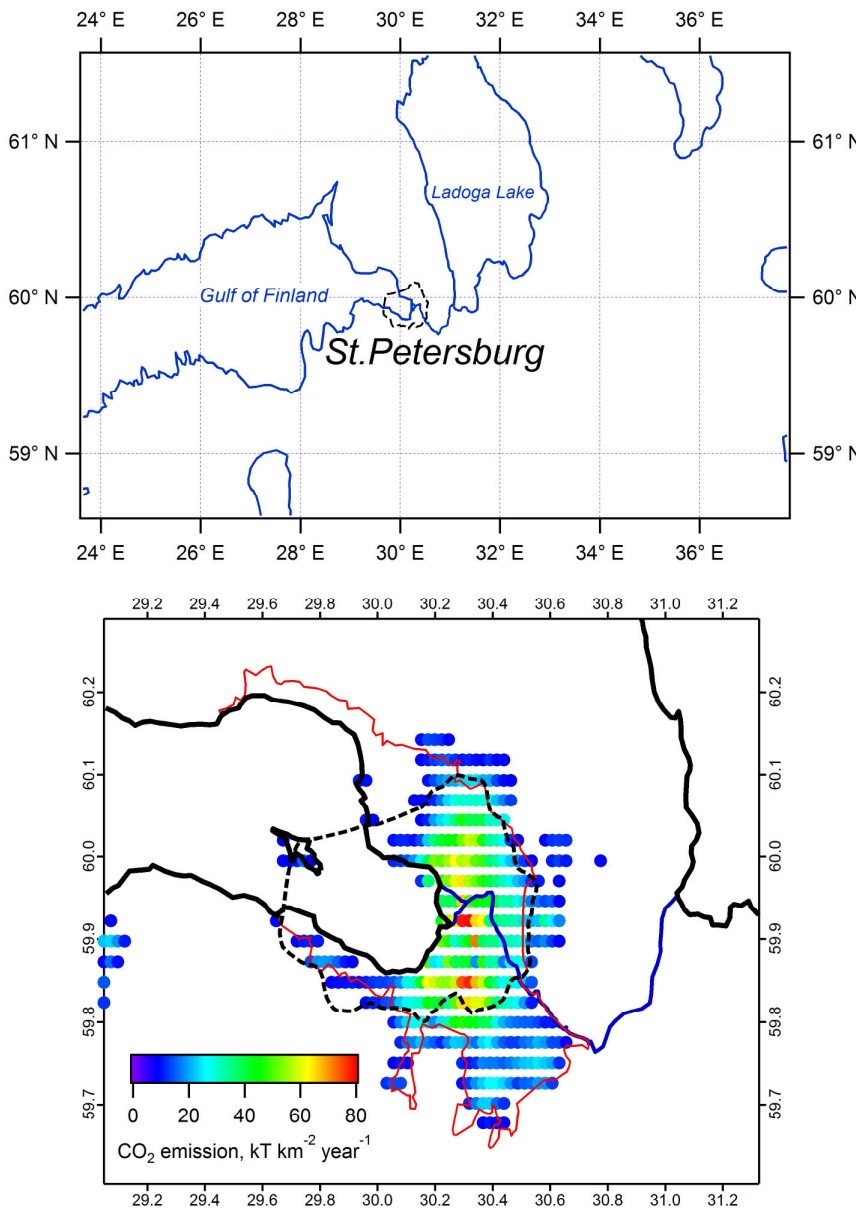

**Figure 3: Top panel: Map of the spatial domain specified in the HYSPLIT model configuration – the city of St. Petersburg and the surrounding area (top image). Bottom panel: The pixel map of the $CO_2$ emissions generated using ODIAC2019.**

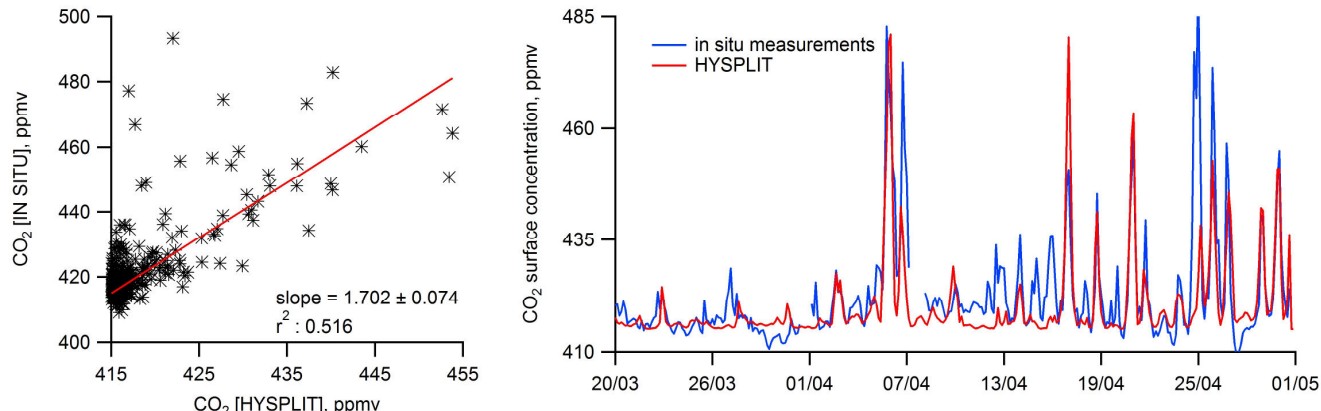

**Figure 4: Comparison of the HYSPLIT simulations and the in situ measurements of surface $CO_2$ concentration in Peterhof (59.88° N, 29.82° E) in March-April 2019. Left panel: The values of surface $CO_2$ compared with the results of HYSPLIT simulations without scaling of the ODIAC emissions data. Right panel: HYSPLIT data obtained using scaled ODIAC $CO_2$ emissions compared with observed surface $CO_2$. Measurement and simulation data are averaged over 3-hour intervals.**

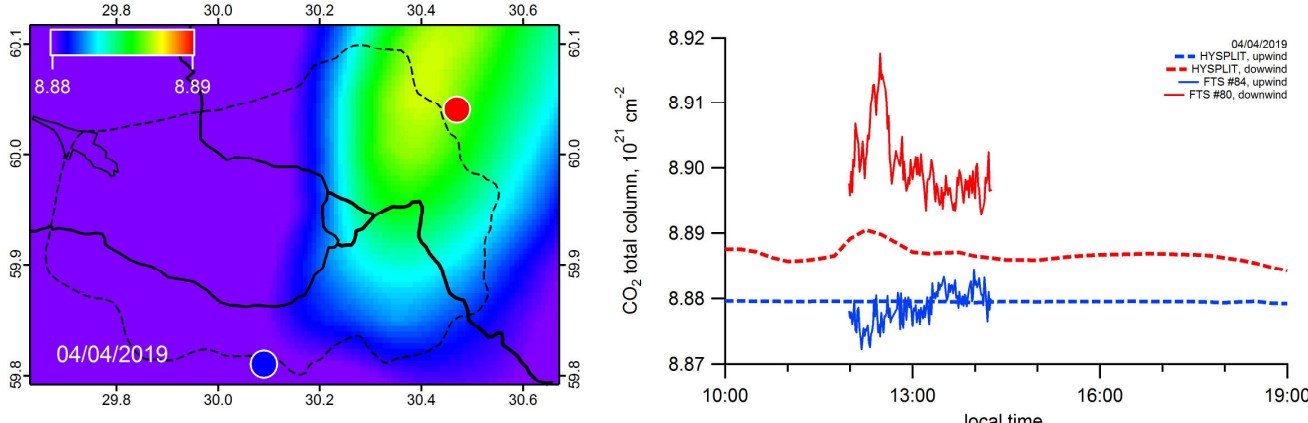

**Figure 5: Left panel: Urban pollution CO₂ plume over St. Petersburg calculated by HYSPLIT model for April 4, 2019 (10:00 UTC). The colour bar designates the CO₂ total column in units $10^{21}$ cm⁻². The blue and red circles indicate the locations of upwind and downwind FTS observations, accordingly. Right panel: Time series of measured (FTS) and simulated (HYSPLIT, without scaling of the ODIAC emissions data) CO₂ total column at the upwind (blue lines) and downwind (red lines) locations for the same day.**

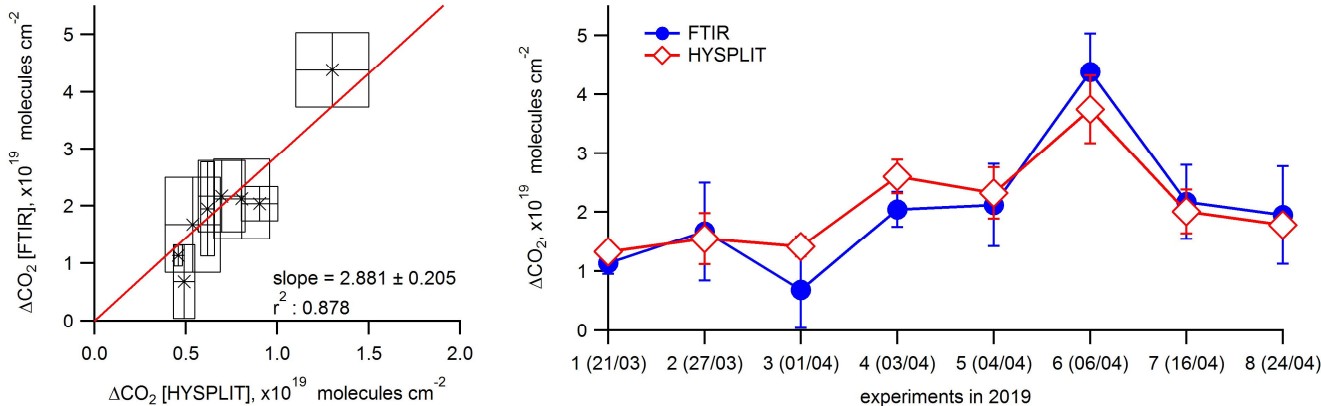

**Figure 6: Left panel: The values of $\Delta CO_2$ (see text) acquired during the field FTIR observations in 2019 compared with the results of HYSPLIT simulations without scaling of the ODIAC data. Measurement and simulation data are averaged over time intervals of FTIR measurements. Right panel: HYSPLIT data obtained using scaled ODIAC $CO_2$ emissions compared with observed $\Delta CO_2$. Dots are connected by lines for illustrative purposes only.**

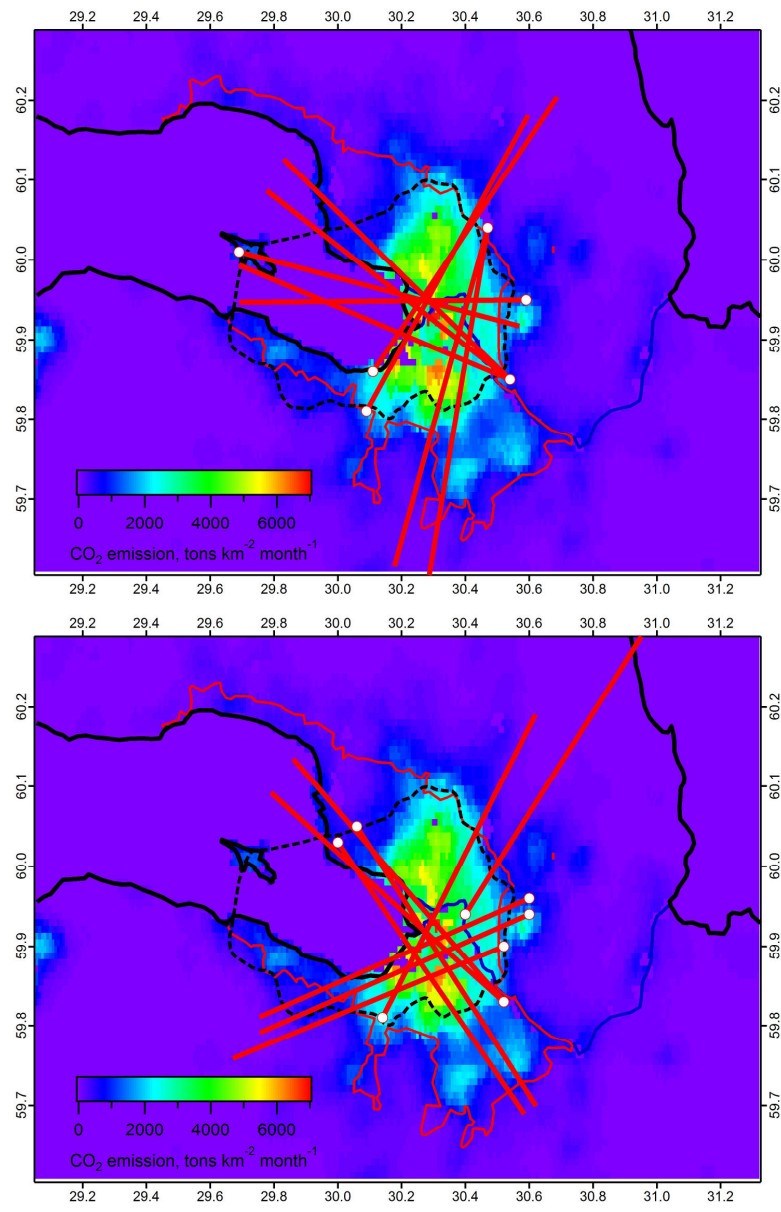

**Figure 7: Map of air mass trajectories corresponding to field measurements of EMME experiments in March-April 2019 (top) and March-April 2020 (bottom). For simplicity, the trajectories are designated by straight lines 50 km long, ending at the locations of downwind FTIR measurements.**

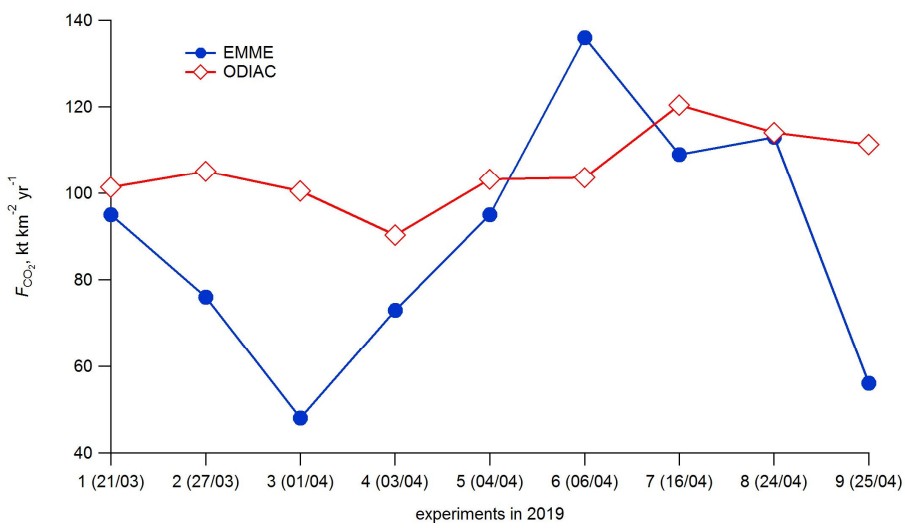

**Figure 8: The CO₂ area flux ($F_{CO_2}$) obtained on the basis of the mass balance approach (EMME-2019) compared to the CO₂ area flux derived from scaled ODIAC data. The calculations are made for the trajectories shown in Fig. 7. Dots are connected by lines for illustrative purposes only.**

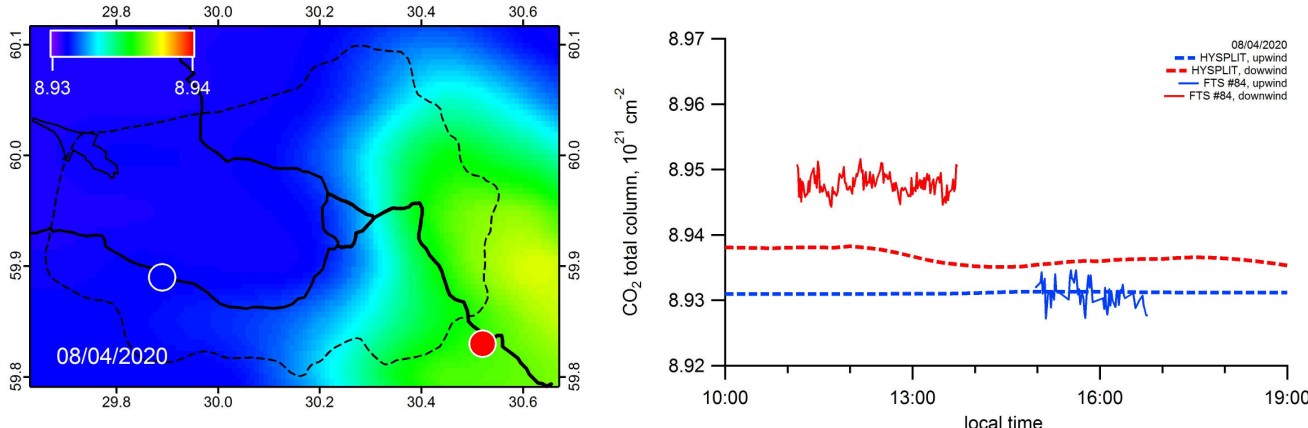

**Figure 9: Left panel: Urban pollution $CO_2$ plume over St. Petersburg calculated by HYSPLIT model for April 8, 2020 (10:00 UTC). The colour bar designates the $CO_2$ total column in units $10^{21}$ cm$^{-2}$. The blue and red circles indicate the locations of upwind and downwind FTS observations, accordingly. Right panel: Time series of measured (FTS) and simulated (HYSPLIT, without scaling of the ODIAC emissions data) $CO_2$ total column at the upwind (blue lines) and downwind (red lines) locations for the same day.**

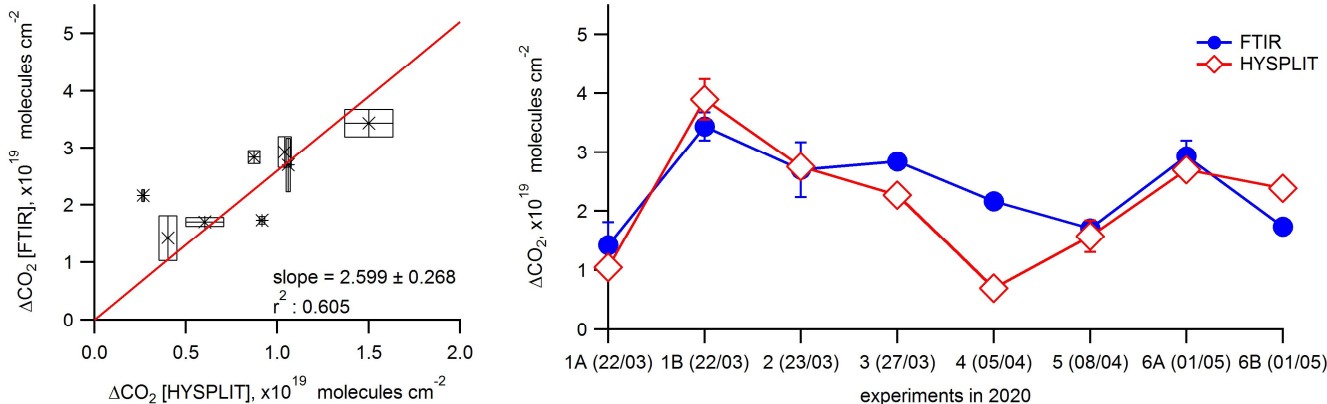

**Figure 10: Left panel: The values of $\Delta CO_2$ (see text) acquired during the field FTIR observations in 2020 compared with the results of HYSPLIT simulations before the process of scaling of the ODIAC data. Measurement and simulation data are averaged over time intervals of FTIR measurements. Right panel: HYSPLIT data obtained using scaled ODIAC CO2 emissions compared with observed $\Delta CO_2$. Dots are connected by lines for illustrative purposes only.**

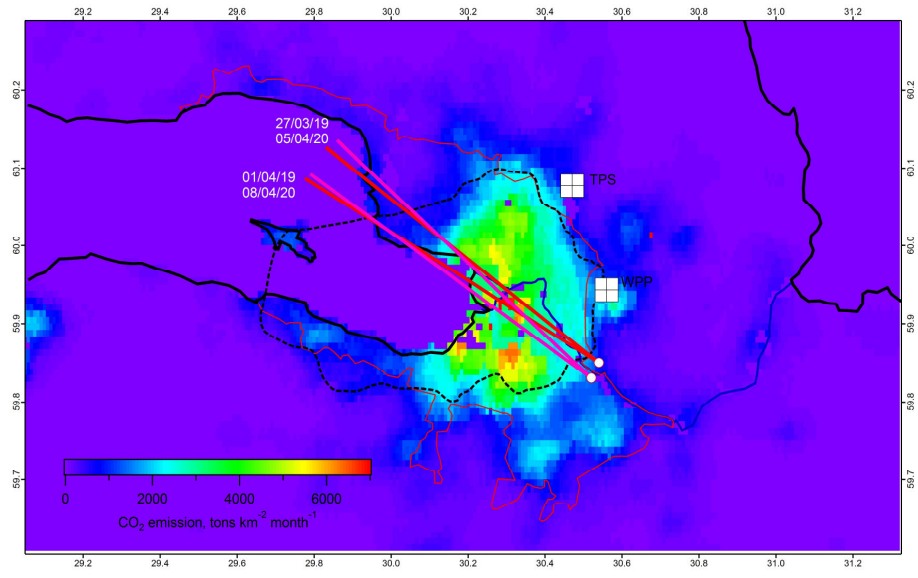

**Figure 11: Map of similar air trajectories and similar downwind measurement locations for EMME-2019/2020 experiments. For simplicity, the trajectories are marked with straight lines 50 km long, ending at the locations of downwind FTIR measurements. The locations of a thermal power station (TPS) on the north-eastern side and a solid waste processing plant (WPP) on the eastern side are also indicated.**

 **Appendix A: Location of ground-based measurement points with respect to the urban pollution plume**

The location of FTS field measurements is shown on the maps of vertically integrated $CO_2$ (total column: TC) produced by HYSPLIT for selected campaign days in 2019 and 2020 (10:00 UTC), see Fig. A1 and A2. The locations of the FTS instruments on the upwind and downwind sides are indicated by blue and red circles, respectively. Note that in 2020 there were days when the downwind measurements were performed twice, at different locations – on March 23 and May 1 (see Fig. A2).

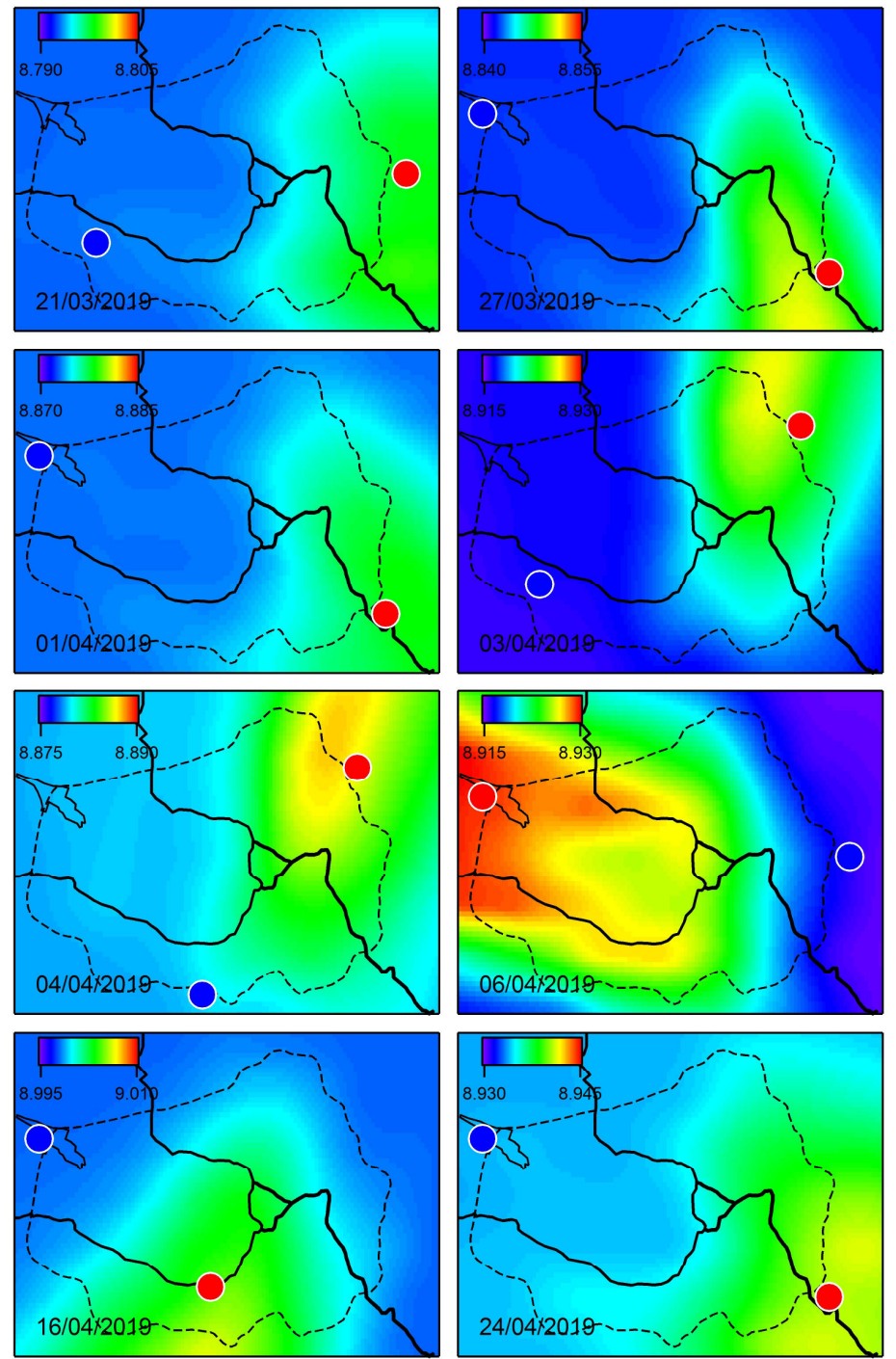

**Figure A1:** Urban pollution $CO_2$ plume over St. Petersburg calculated with HYSPLIT model for the days of field campaign in 2019 (10:00 UTC). The colour bar units for $TC_{CO2}$ are $10^{21}$ cm$^{-2}$. The blue and red circles indicate the locations of upwind and downwind FTS observations, accordingly.

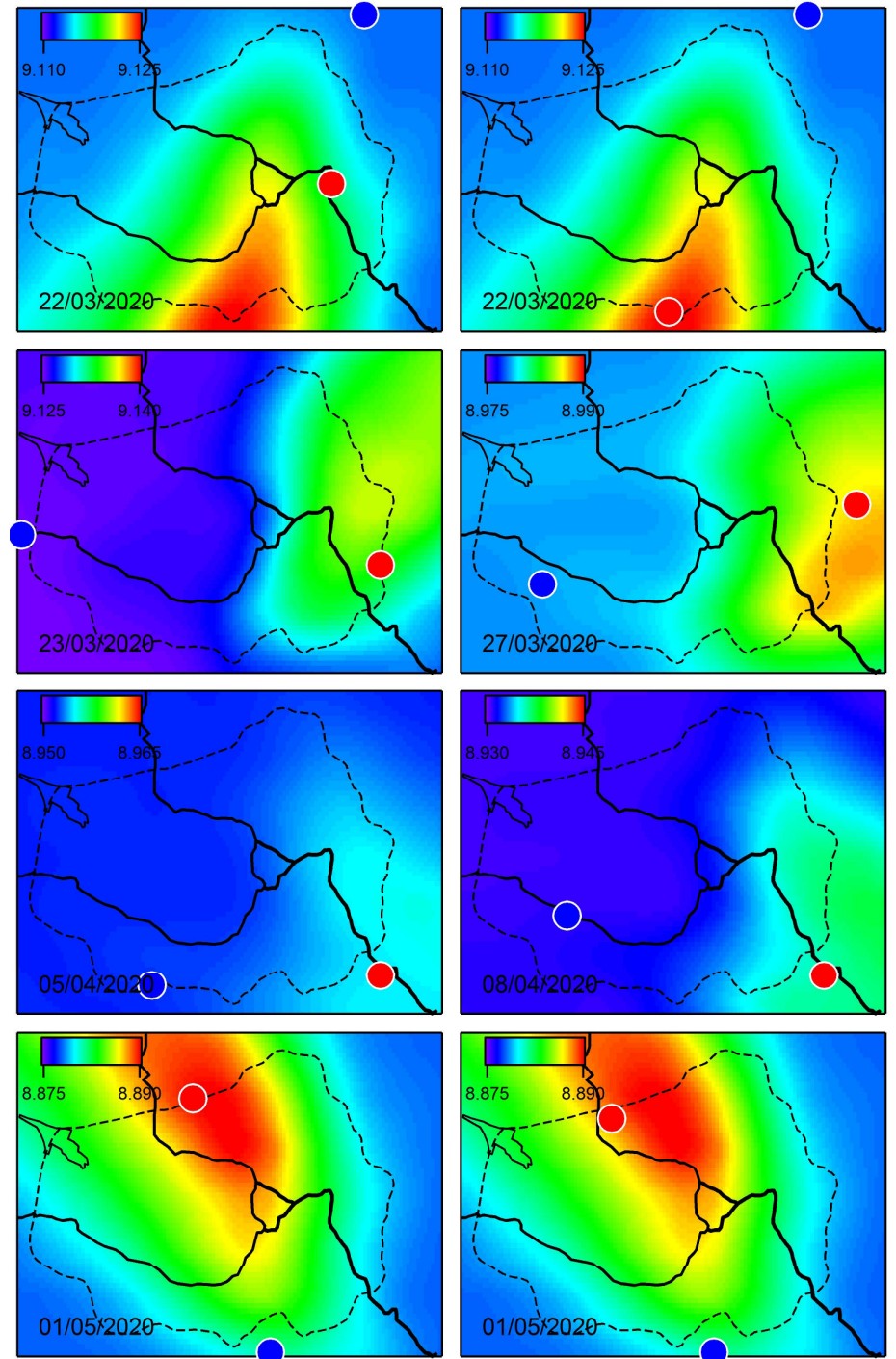

**Figure A2: Urban pollution CO$_2$ plume over St. Petersburg calculated with HYSPLIT model for the days of field campaign in 2020 (10:00 UTC). The colour bar units for TC$_{CO2}$ are $10^{21}$ cm$^{-2}$. The blue and red circles indicate the locations of upwind and downwind FTS observations, accordingly.**

**Appendix B: The data series of measured and calculated CO$_2$ content**

The upwind and downwind CO$_2$ total column values acquired from FTIR measurements and HYSPLIT calculations are shown for selected campaign days in 2019 and 2020 in Fig. B1 and B2. The HYSPLIT data are in fact the values of an integrated vertical column in the range of 0-1500 meters (10 altitude layers) calculated with the 15-minute time step. The background level of the CO$_2$ column is set equal to an average of the FTIR upwind measurements during a day. Note that in 2020, there were days when the downwind measurements were performed twice, at different locations – on March 22 and

May 1 (see Fig. B2).

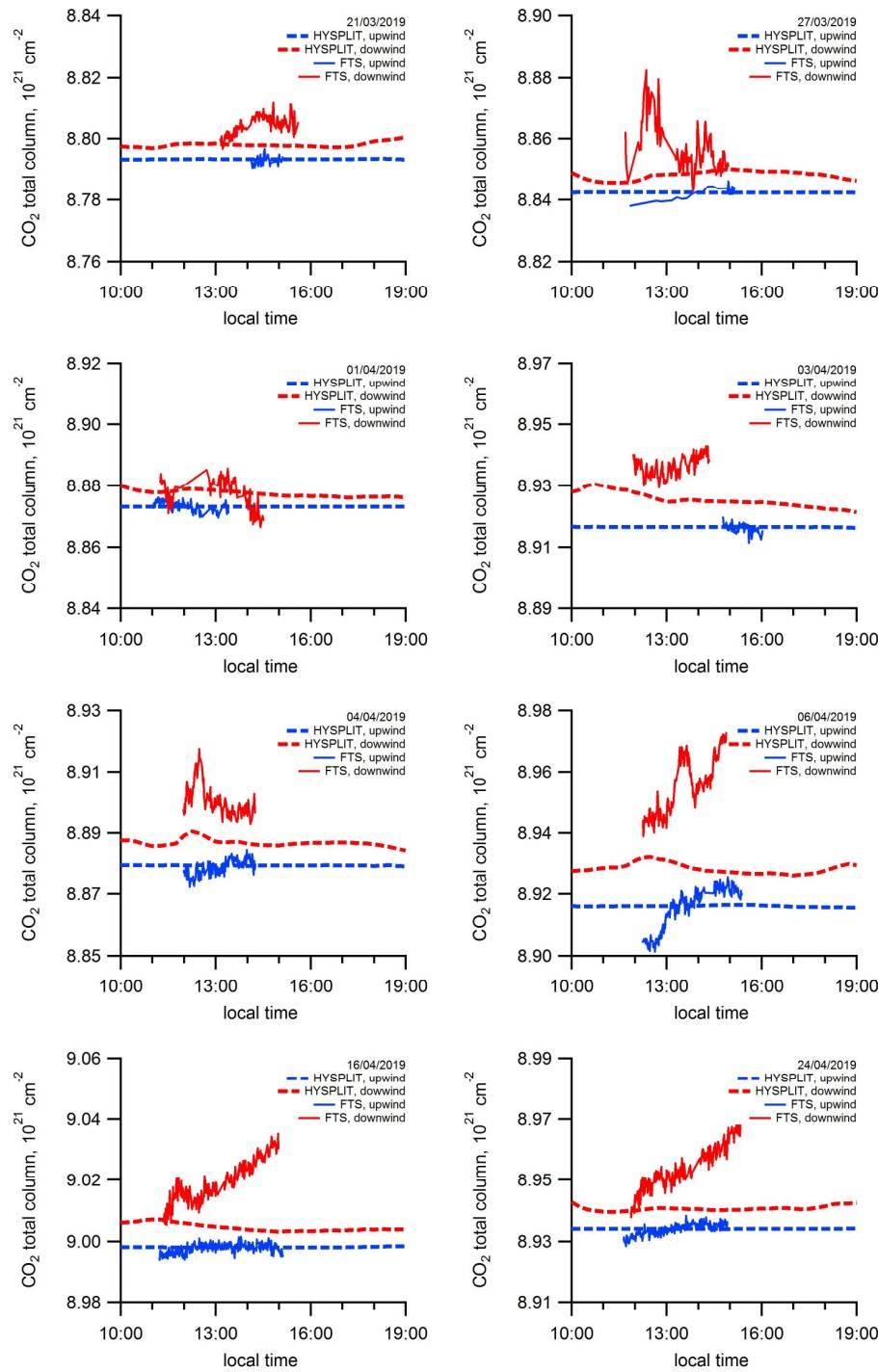

**Figure B1: Time series of measured (FTS) and simulated (HYSPLIT, without scaling of the ODIAC emissions data) CO$_2$ total column at the upwind (blue lines) and downwind (red lines) locations for selected campaign days in 2019.**

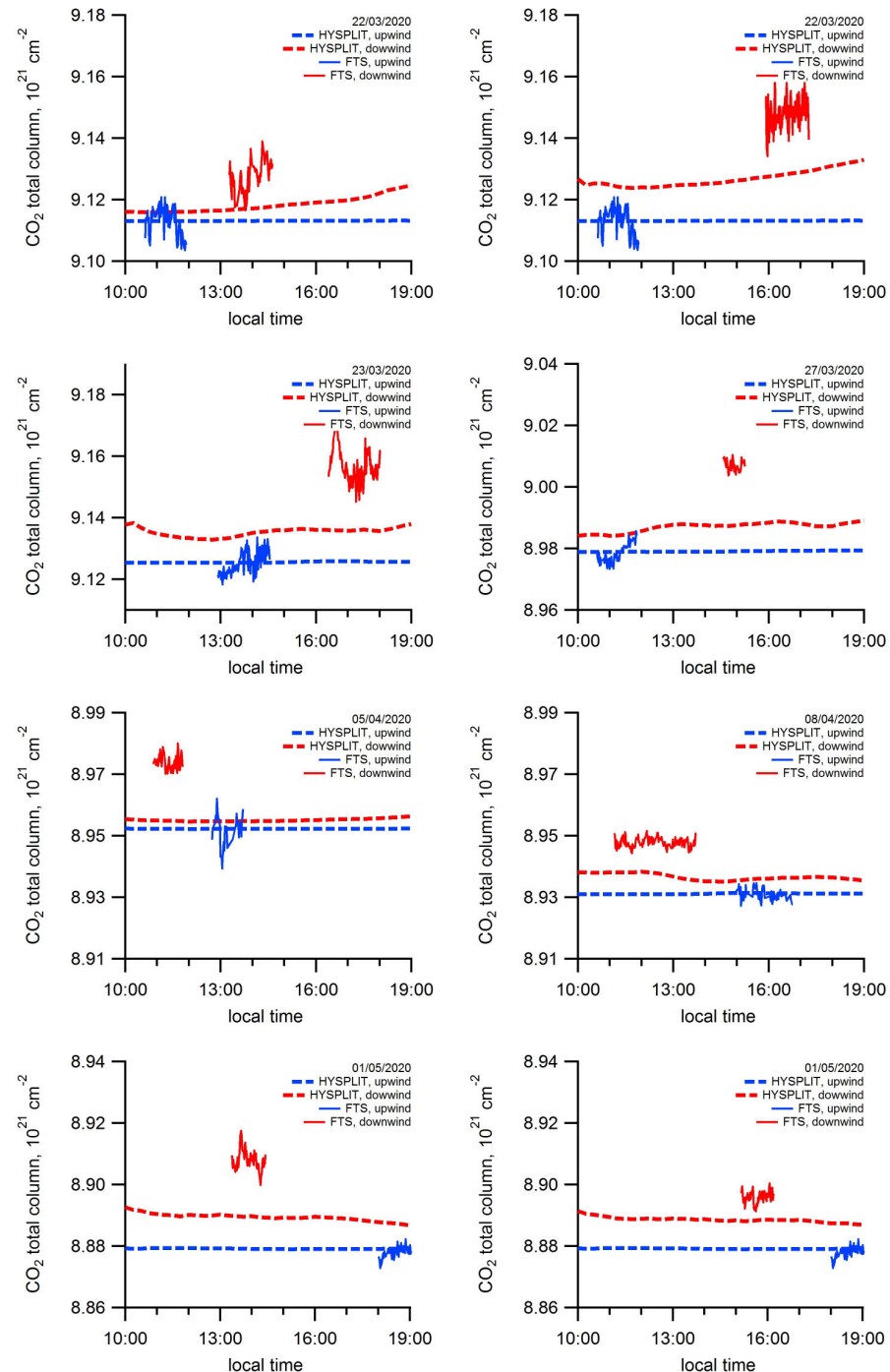

**Figure B2: Time series of measured (FTS) and simulated (HYSPLIT, without scaling of the ODIAC emissions data) CO₂ total column at the upwind (blue lines) and downwind (red lines) locations for selected campaign days in 2020.**