# Peer review of "The CO2 integral emission by the megacity of St. Petersburg as quantified from ground-based FTIR measurements combined with dispersion modelling"

_Atmospheric Chemistry and Physics, 2020_

## Author Comment (AC1)

Authors' reply to referee #1

We express our gratitude to the referee for reviewing our paper and for the valuable corrections and suggestions to improve our work. As an outcome, we have carefully revised the manuscript, with a number of modifications being done in compliance with all the raised issues.

Below, the actual comments of the referee are given in **`bold courier font and blue colour`**.
The text added to the revised version of the manuscript is marked by red colour.

**`*General structural comments:`**

**`*1. There are no clear distinction between the methods, results and discussion sections in the manuscript. The manuscript should be reorganized in order to better guide the audience through the entire analysis step by step.`**

In response to the referee's comment, we restructured our article as follows:

    1 Introduction
    2 Methods and instrumentation
    2.1 Bruker EM27/SUN FTS and spectra processing
    2.2 A priori data on FF $CO_2$ emissions
    2.3 The HYSPLIT model general setup
    2.4 Test simulations of ground-level $CO_2$ concentrations
    3 Evaluation of integrated $CO_2$ emissions from field FTIR measurements
    3.1 The results of the EMME-2019 campaign
    3.2 The results of EMME-2020 and comparison with EMME-2019
    4 Summary and conclusions

**`*2. The introduction doesn't clearly outline the scope of the work. The bullet points at the end need to be expanded in more details. All the subsequent sections and subsections that follow the introduction should be outlined in the introduction in the same order.`**

In response to the referee's comment, we have rewritten this of introduction as follows:

    The present study continues the analysis of the data of EMME-2019 and demonstrates the first results of the 2020 campaign. We concentrate our efforts on the $CO_2$ emissions leaving the results relevant to other gases beyond the scope of the research. As an extension to the work by Makarova et al. (2021) our goal in this paper is to estimate the integral $CO_2$ emission by St. Petersburg megacity rather than area fluxes. Completing this task consists of the following basic steps:
- We use mobile FTIR measurements to obtain $CO_2$ column enhancements ($\Delta CO_2$) related to urban anthropogenic emissions.
- We adapt the ODIAC database (Oda and Maksyutov, 2011) to construct a priori information on the spatio-temporal distribution of anthropogenic $CO_2$ emissions on the territory of St. Petersburg.
- We initialize the HYSPLIT dispersion model, HYbrid Single-Particle Lagrangian Integrated Trajectories (Draxler and Hess, 1998; Stein et al., 2015) with the ODIAC emissions to simulate $CO_2$ 3D field over the city of St. Petersburg.
- We evaluate the performance of our HYSPLIT model setup by calculating the surface $CO_2$

concentrations and comparing them with the routine in-situ measurement results (Foka et al., 2019).

- We scale the emission input data for the HYSPLIT model simulations in order to reproduce the observed $\Delta CO_2$.

- Finally, from the scaled emission a priori data we get the estimate of integral $CO_2$ emission by St.Petersburg.

The paper is organized as follows. Section 2 describes the methods and instruments, including a description of the EMME measurement campaign and the equipment used, methods for processing the measurement results, the configuration of the HYSPLIT model and its evaluation based on calculations of ground-level $CO_2$ concentrations. Section 3 presents main results of EMME-2019 and EMME-2020 including estimates of integrated $CO_2$ emissions derived from FTIR measurements of the 2019 and 2020 field campaigns, combined with HYSPLIT model simulations. Section 4 contains a summary of our findings.

**\*3. The authors should not assume that the readers are familiar with the previous work neither they should expect them to read other manuscripts to understand the scope or the methods used in the current study. While citation to the earlier study ( Makarova et al. (2020) ) is encouraged in moderation, methods should be briefly explained in the current manuscript as well.**

We have added more information about the 2019 and 2020 field measurements to the new Section 2 "Methods and instrumentation".

**\*Scientific comments:**

**\*1. It is mentioned that during the 2019 campaign two FTIR instruments were used whereas only one instrument was deployed during the 2020 campaign. There are two questions regarding this: 1) Where were the locations of the two instruments in 2019? If there's more than one configuration, clearly outline the locations at each specific day. Presenting the measurement locations on the map is encouraged. 2) How did you estimate $\Delta CO2$ in 2020 by using only one instrument. Did you move the instrument within one day? If yes, what time? Outline clearly maybe in a table.**

The strategy of our mobile experiments was to conduct paired series of measurements – at the upwind and downwind sides of the city. This was noted in the text of the article at the beginning of Section 2 in the original manuscript version (*"The EMME measurement campaign (short summary)"*): "*two mobile EM27/SUN FTIR spectrometers were used in the field experiment for simultaneous measurements inside and outside of the air pollution plume*" (in 2019) and "*one spectrometer which was transported between clean and polluted locations within one day*" (in 2020). Following the reviewer's comments, we expanded this part of the text and included the tables with information about each mobile experiment in 2019 and 2020 (see below, beginning of a new Section 2 in the revised manuscript, "Methods and instrumentation"):

[revised manuscript text omitted]

*You should keep in mind that due to the air mass dependencies the diurnal differences do not necessarily reflect changes in the emissions. So knowing the measurement times is critical.

We agree with the referee's comment. Indeed, TCCON and COCCON FTIR observations have high precision and accuracy and usually reveal the $CO_2$ diurnal cycle with the maximum around the local noon. For our case, it is assumed that the $CO_2$ diurnal cycle for the background atmospheric conditions could be estimated using FTIR observations at the upwind location where the influence of air pollution is minimal. According to the results obtained during 2019 campaign, the pronounced diurnal variations of $CO_2$ for upwind locations were observed during only one day (06.04.2019, please see Appendix B, Figure B1). For this day the $CO_2$ TC had the variation of 0.2%. For other seven days of EMME-2019, the $CO_2$ diurnal cycle showed significantly lower (or even negligible) amplitudes (see Figure B1). Therefore, relying on the results of mobile campaign of 2019 we assumed that it is quite possible to perform mobile FTIR observations in 2020 using only one EM27/SUN spectrometer (i.e. making FTIR measurements at up- and downwind locations sequentially) even if this approach can introduce certain additional errors.

**\*2. Which retrieval algorithm was used for analyzing the FTIR spectra?**

To analyze the FTIR spectra we used the free software PROFFAST specially developed for COCCON network. We explain this in the new subsection 2.1 "Bruker EM27/SUN FTS and spectra processing" of Section 2 "Methods and instrumentation" that we add to the text:

To process the spectral data we used the free software PROFFAST which had been specially developed for COCCON network (COCCON, 2021; Frey et al., 2019). PROFFAST has been developed by KIT in the framework of several ESA projects for processing the raw data delivered by the EM27/SUN FTS. For the retrievals of total columns (TCs) of target species the following spectral bands are used (Frey et al., 2015; Hase et al., 2016; Frey et al., 2019): 4210-4320 $cm^{-1}$ (target gas – CO, interfering gases – $H_2O$, HDO, $CH_4$), 5897-6145 $cm^{-1}$ (target gas – $CH_4$, interfering gases – $H_2O$, HDO, $CO_2$), 6173-6390 $cm^{-1}$ (target gas – $CO_2$, interfering gases – $H_2O$, HDO, $CH_4$), 7765-8005 $cm^{-1}$ (target gas – $O_2$, interfering gases – $H_2O$, HF, $CO_2$), and 8353-8463 $cm^{-1}$ (target gas – $H_2O$). The retrieval algorithm requires the following input: temperature profile in the atmosphere, pressure at the ground level, and the a priori data on the mole fraction vertical distribution of the atmospheric trace gases. These data are generated by the TCCON network software which ensures their compatibility over the TCCON network (TCCON, 2021). The close agreement of EM27/SUN observations analyzed with PROFFAST with a collocated TCCON spectrometer has been demonstrated in the framework of the ESA project FRM4GHG (Sha, 2020).

**\*3. Could you show the timeseries of $XCO_2$ or $\Delta CO_2$ showing FTIR measurements for a typical day or the entire period of measurements? This would help the readers understand the daily variations and enhancements better.**

We followed the advise of the referee and now the revised version of the manuscript contains an appendix in which the data series of measured and calculated $CO_2$ content are shown for the selected campaign days in 2019 and 2020 (Appendix B, Fig. B1 and B2):

**Appendix B: The data series of measured and calculated $CO_2$ content**

The upwind and downwind $CO_2$ total column values acquired from FTIR measurements and HYSPLIT calculations are shown for selected campaign days in 2019 and 2020 in Fig. B1 and B2. The HYSPLIT data are in fact the values of an integrated vertical column in the range of 0-1500 meters (10 altitude layers) calculated with the 15-minute time step. The background level of the $CO_2$ column is set equal to an average of the FTIR upwind measurements during a day. Note

that in 2020, there were days when the downwind measurements were performed twice, at different locations – on March 22 and May 1 (see Fig. B2).

[Figure]

**Figure B1: Time series of measured (FTS) and simulated (HYSPLIT, without scaling of the ODIAC emissions data) $CO_2$ total column at the upwind (blue lines) and downwind (red lines) locations for selected campaign days in 2019.**

[Figure]

**Figure B2: Time series of measured (FTS) and simulated (HYSPLIT, without scaling of the ODIAC emissions data) $CO_2$ total column at the upwind (blue lines) and downwind (red lines) locations for selected campaign days in 2020.**

In addition, two examples of HYSPLIT-simulated $CO_2$ plumes and corresponding time series of $CO_2$ total column measurements for the typical days of experiments in 2019 and 2020 are presented in Figs. 5 and 9:

[Figure]

**Figure 5: Left panel: Urban pollution CO$_2$ plume over St. Petersburg calculated by HYSPLIT model for April 4, 2019 (10:00 UTC). The colour bar designates the CO$_2$ total column in units 10$^{21}$ cm$^{-2}$. The blue and red circles indicate the locations of upwind and downwind FTS observations, accordingly. Right panel: Time series of measured (FTS) and simulated (HYSPLIT, without scaling of the ODIAC emissions data) CO$_2$ total column at the upwind (blue lines) and downwind (red lines) locations for the same day.**

[Figure]

**Figure 9: Left panel: Urban pollution CO$_2$ plume over St. Petersburg calculated by HYSPLIT model for April 8, 2020 (10:00 UTC). The colour bar designates the CO$_2$ total column in units 10$^{21}$ cm$^{-2}$. The blue and red circles indicate the locations of upwind and downwind FTS observations, accordingly. Right panel: Time series of measured (FTS) and simulated (HYSPLIT, without scaling of the ODIAC emissions data) CO$_2$ total column at the upwind (blue lines) and downwind (red lines) locations for the same day.**

**\*4. How the two FTIR instruments compare to each other when measuring side by side? How are you accounting for potential instrument biases?**

Yes, we certainly took care of this issue. To give some details we have added the following text to subsection 2.1 "2.1 Bruker EM27/SUN FTS and spectra processing":

In order to obtain a reliable value of the $CO_2$ emission for St. Petersburg, it is necessary to eliminate possible systematic error caused by the instrument bias. This goal was reached by carrying out a cross-calibration of the instruments. In April-May 2019 both instruments passed a four day cross-calibration. The comparison of side-by-side measurements of $X_{CO2}$ by FTS#80 and FTS#84 allowed determining calibration factor which was used for converting $X_{CO2}$ measured by FTS#80 to the scale of FTS#84. Detailed information about side-by-side calibration of FTIR-spectrometers is given in the paper by Makarova et al. (2021).

**\*5. You mentioned you have excluded a pixel from ODIAC priors because the emissions seemed to be an outlier. What if it's the emissions from a power plant(s) that was misattributed to a wrong location? It is expected that industrial sources and power plants stand out as they are point sources with significant amount of emissions. If you any valid reason to modify emissions from that pixel you should clearly state that you are not using ODIAC but a modified version of ODIAC.**

Indeed, power plants and industrial enterprises normally manifest themselves as point sources of strong emission. However, we cannot confidently attribute this particular ODIAC pixel to any source of this type, since there is no such object near it. There are about a dozen of large thermal power plants on the territory of St. Petersburg, but all of them appear to be rather far from this location. Given the reviewer's doubts, we have studied this issue more thoroughly. Despite the lack of published data on anthropogenic $CO_2$ emissions at the city scale, we were able to explore detailed reports from municipal inventories of stationary air pollution sources (unpublished, but available on request). According to the inventories of $NO_x$, CO, $SO_2$, $NH_3$, VOC and PM10 pollutants, there are no stationary objects of an extreme emission close to the point of our interest. Thus, we feel confident to smooth out this outlier when we configure emissions to run the HYSPLIT model. However, we do not consider it necessary to mark this as an "ODIAC modification". Removing one pixel from several thousand in the database does not mean that it is modified significantly. In any case, it is not possible to implement ODIAC directly into HYSPLIT, without adaptation. Nevertheless, we have added a few words of justification to the appropriate place in the text:

Normally, power plants and industrial enterprises manifest themselves as point sources of strong emission. However, we cannot confidently attribute this particular ODIAC pixel to any source of this type, since there is no such object near it. There are about a dozen of large thermal power plants on the territory of St. Petersburg, but all of them appear to be rather far from this location. Despite the lack of published data on anthropogenic $CO_2$ emissions at the city scale, we were able to explore detailed reports from municipal inventories of stationary air pollution sources (unpublished, but available on request). According to the inventories of $NO_x$, CO, $SO_2$, $NH_3$, VOC and PM10 pollutants, there are no stationary objects of an extreme emission close to the point of our interest. Thus we feel confident to smooth out this outlier and replace it by the value averaged over the neighboring ODIAC pixels.

**\*6. For comparisons of in-situ measurements and HYSPLIT model results you are using a fixed background of 415 ppm. Given the day to day variations in the $CO_2$ levels it's not very reasonable to use a fixed value for the entire period of the analysis unless you bring proof that this was the case for St Petersburg during the campaign.**

As it was mentioned in the article (subsection 3.3 in the original manuscript version, *"Simulations of ground-level $CO_2$ concentrations"*), routine measurements of the $CO_2$ surface concentrations have been carried out since 2013 using a gas analyzer Los Gatos Research GGA 24r-EP located at the

atmospheric monitoring station of Saint-Petersburg State University in Peterhof (59.88°N, 29.83°E). The instrument is installed on the outskirts of a small town of Peterhof in the suburbs of St. Petersburg (see location in Fig. 1). This place is far enough away from busy streets and other local sources of pollution, with an ambient air intake being 3 meters above the surface. Thus, for testing HYSPLIT model setup for the St. Petersburg region, we used background value of the $CO_2$ surface concentration near Peterhof during the 2019 EMME measurement campaign – from March 20 to April 30, 2019 (Makarova et al., 2021). We used the background value of the surface concentration equal to $415 \pm 2$ ppmv, which is the mean value of diurnal minima during the campaign. We have added this explanation to the revised manuscript (subsection 2.4 "Test simulations of ground-level $CO_2$ concentrations"):

The background value of the surface concentration is taken as 415 ppmv based on the local measurements ($415 \pm 2$ ppmv is the mean value of diurnal minima during the campaign from March 20 to April 30, 2019).

**\*7. It will be useful if you also plot the one on one curve for HYSPLIT vs in-situ measurements from which you found out a scaling factor.**

We have changed Figure 4 to show the regression plot of HYSPLIT vs in-situ $CO_2$ measurements, as suggested by the referee.

[Figure]

**Figure 4: Comparison of the HYSPLIT simulations and the in situ measurements of surface $CO_2$ concentration in Peterhof (59.88° N, 29.82° E) in March-April 2019. Left panel: The values of surface $CO_2$ compared with the results of HYSPLIT simulations without scaling of the ODIAC emissions data. Right panel: HYSPLIT data obtained using scaled ODIAC $CO_2$ emissions compared with observed surface $CO_2$. Measurement and simulation data are averaged over 3-hour intervals.**

**\*8. Please describe in details how did you estimate fluxes using the mass balance method. Bring equations if necessary.**

In fact, calculations, which use the mass balance approach, are only supplementary in the present paper. It was the subject of our earlier study published by Makarova et al. (2021) with all the necessary equations presented there. The details of the flux estimates obtained by mass balance approach are definitely out of the scope of the current study. However, we have added the following brief explanations to the revised text of our manuscript:

Basically, the mass balance approach was adopted in the form of a one-box model, and the area flux $F$ was calculated using the following equation:

$$F = \frac{\Delta CO_2 \cdot V}{L}$$

(2)

where $F$ is the $CO_2$ area flux, $\Delta CO_2$ is the difference between the downwind and upwind FTIR measurements, $V$ is the mean wind speed and $L$ is the mean path length of an air parcel which goes through the urban area (Chen et al., 2016).

**\*9. How are you computing column ΔCO2 using HYSPLIT? Do you take an integral over the vertical layers of 1m-1500m?**

Exactly so. We take an integral over the $CO_2$ concentrations calculated using HYSPLIT for 10 vertical layers in the range of 0-1500 meters. This is what we write in the original text of our manuscript (beginning of Section 4 in the original manuscript version, "*Evaluation of integrated CO$_2$ emissions from field FTIR measurements*"): "*HYSPLIT model was configured to simulate CO$_2$ concentrations at 10 altitude levels (0-1500 m), which were then integrated to obtain the CO$_2$ column in the boundary layer. The differences between the results of FTIR measurements of the CO$_2$ TC inside and outside the pollution plume (ΔCO$_2$) were compared with the differences in the CO$_2$ column in the boundary layer simulated by HYSPLIT at the corresponding locations*".

**\*Detailed comments:**

**\*Line 26: From the value in brackets I imply that emission estimates during the lockdown period was higher than the rest but the last sentence of abstract is suggesting there was a 8% decrease. Please clarify!**

Indeed, this part of the text is not entirely clear. In fact, the strict allocation of the period of influence of the quarantine based on the formal terms of its introduction is not quite correct, since some decline in business activity could have started even before the official introduction of restrictive measures. Therefore, in the revised version of our manuscript, we abandoned this approach and now compare the results of all measurements in 2020 with the results of all measurements in 2019. We have made the appropriate changes to the Annotation and to Section 4 "Summary and conclusions":

The most important finding of the present study based on the analysis of two observational campaigns is a significantly higher $CO_2$ emission from the megacity of St. Petersburg as compared to the data of municipal inventory: ~75800±5400 kt yr$^{-1}$ for 2019, ~68400±7100 kt yr$^{-1}$ for 2020 versus ~30000 kt yr$^{-1}$ reported by official inventory. The comparison of the $CO_2$ emissions obtained during the COVID-19 lockdown period in 2020 to the results obtained during the same period of 2019 demonstrated the decrease in emission of 10% or 7400 kt yr$^{-1}$.

**\*Line 42: Please consider adding a proper citation.**

The following reference has been added: UN: United Nations, Department of Economic and Social Affairs, Population Division: World Urbanization Prospects 2018: Highlights. https://population.un.org/wup/Publications/Files/WUP2018-Highlights.pdf, last access 29 March 2021.

**\*Line 43: Do you mean anthropogenic CO$_2$ emissions? Also having total fossil fuel consumption is not enough to estimate anthropogenic emissions. There are emissions**

associated with land use change, agriculture and other industries such as cement production.

In response to this comment of the reviewer, we have rewritten the mentioned sentence:

The vast majority of anthropogenic $CO_2$ emissions in developed countries are associated with the burning of fossil fuels (FF) and can be estimated with good accuracy on the basis of the total fuel consumption.

**\*Line 77 and 78: Please bring proper citations for both official and unofficial population estimates.**

The following reference containing the official information on the population of St. Petersburg has been added:

St. Petersburg Center for Information and Analytics: Report on Demographic monitoring in St. Petersburg: Q3 2020, https://www.gov.spb.ru/helper/new_stat/, 2020, (in Russian).

The reference containing the unofficial estimates of the population of St. Petersburg has been added:

Shevlyagina M. The real population of St. Petersburg exceeds 7 million people, https://spbdnevnik.ru/news/2020-02-27/realnoe-naselenie-peterburga-prevyshaet-7-millionov-chelovek, last access 29 March 2021 (in Russian).

**\*Line 88: How much is the contribution from transport in St Petersburg? It's important since you are mentioning transport emission changes during lock down period later in the text.**

As we noted later in the text of the manuscript (subsection 4.2 in the original manuscript version, *"The results of EMME-2020 and comparison with EMME-2019"*), official data on the contribution of transport to the total $CO_2$ emissions in St. Petersburg are not available:

*Since we have no official data on the $CO_2$ emissions by traffic at our disposal, we used the average estimate for European countries, according to which the contribution of traffic to total emission constitutes 30% (European Parliament News, 2020).*

According to the report prepared by the analytical center of Gazprom public joint stock company (Gazprom, 2021) the total $CO_2$ emission from motor vehicles in St. Petersburg was of ~10 Mt/year in 2014. This value is about one third of the total $CO_2$ emission of St.Petersburg (~30 Mt/year) reported by the municipal Environmental Committee (Serebritsky, 2018). This supports our assumption of a 30% contribution of transport to total $CO_2$ emissions.

Gazprom: The ecological effect of the conversion of motor vehicles to gas-powered fuel in the regions of the Russian Federation, https://www.gazprom.ru/f/posts/22/538143/atlas-ecology-effect-gas-transport.pdf, last access 30 March 2021 (in Russian).

Serebritsky, I.A., (Ed.): The Report on Environmental Conditions in St. Petersburg for 2017, https://www.gov.spb.ru/static/writable/ckeditor/uploads/2018/06/29/Doklad_EKOLOGIA2018.pdf, 2018 (in Russian).

**\*Line 113: As mentioned earlier please briefly describe the campaign in a separate paragraph don't assume audience have read the other paper. Then state what are the**

**differences/additions that you are making in this study compared to the earlier one. Also it might be useful to dedicate a separate paragraph about the instrument. Briefly explain how it works and how spectra are retrieved. Better to move this part to the methods section (that doesn't exist at the moment).**

Following the reviewer's suggestion, the new Section 2 of the revised manuscript ("Methods and instrumentation") contains the subsection 2.1 "Bruker EM27/SUN FTS and spectra processing" which describes the instrument and spectra retrieval in detail:

Bruker EM27/SUN (Gisi et al., 2012; Frey et al., 2015, Hase et al., 2016) is a portable robust FTS having low spectral resolution of 0.5 cm$^{-1}$. It was designed for accurate and precise ground-based observations of $CO_2$, $CH_4$ and CO column-averaged abundances ($X_{CO2}$, $X_{CH4}$ and $X_{CO}$) in the atmosphere. These FTIR spectrometers were used to build the COCCON network (COCCON, 2021; Frey et al., 2019). EM27/SUN is equipped with a Camtracker, a solar tracking system developed by KIT (Gisi et al., 2011). A Camtracker consists of an altazimuthal solar tracker, a USB digital camera and "CamTrack" software which processes an image acquired by a camera and controls the tracker's movement. EM27/SUN FTS is designed on the basis of a robust RockSolid™ interferometer having high thermal and vibrational stability; the detailed description of the instrument is given by Gisi et al. (2012). Therefore, this type of instruments is being successfully implemented for setting up fully automated stationary city network MUCCnet (Munich Urban Carbon Column network, Dietrich et al., 2021) and for performing a number of mobile campaigns (Klappenbach et al., 2015; Luther et al., 2019; Makarova et al., 2021).

In our study, we used the dual-channel EM27/SUN with quartz beamsplitter. Additionally, two detectors allow observing $X_{CO}$ and future improvements of the $X_{CO2}$ retrieval (Hase et al., 2016). FTS registers an interferogram which is the Fourier transform of the infrared spectrum of direct solar radiation. The processing of data acquired by EM27/SUN spectrometer consists of the following stages:
- deriving spectra from raw interferograms including a DC-correction and quality assurance procedures (Keppel-Aleks et al., 2007);
- deriving $O_2$, $CO_2$, CO, $H_2O$, and $CH_4$ total columns (TCs) from FTIR spectra by scaling a priori profiles of retrieved gases (Frey et al., 2019; COCCON, 2021).

To process the spectral data we used the free software PROFFAST which had been specially developed for COCCON network (COCCON, 2021; Frey et al., 2019). PROFFAST has been developed by KIT in the framework of several ESA projects for processing the raw data delivered by the EM27/SUN FTS. For the retrievals of total columns (TCs) of target species the following spectral bands are used (Frey et al., 2015; Hase et al., 2016; Frey et al., 2019): 4210-4320 cm$^{-1}$ (target gas – CO, interfering gases – $H_2O$, HDO, $CH_4$), 5897-6145 cm$^{-1}$ (target gas – $CH_4$, interfering gases – $H_2O$, HDO, $CO_2$), 6173-6390 cm$^{-1}$ (target gas – $CO_2$, interfering gases – $H_2O$, HDO, $CH_4$), 7765-8005 cm$^{-1}$ (target gas – $O_2$, interfering gases – $H_2O$, HF, $CO_2$), and 8353-8463 cm$^{-1}$ (target gas – $H_2O$). The retrieval algorithm requires the following input: temperature profile in the atmosphere, pressure at the ground level, and the a priori data on the mole fraction vertical distribution of the atmospheric trace gases. These data are generated by the TCCON network software which ensures their compatibility over the TCCON network (TCCON, 2021). The close agreement of EM27/SUN observations analyzed with PROFFAST with a collocated TCCON spectrometer has been demonstrated in the framework of the ESA project FRM4GHG (Sha, 2020).

**\*Line 150: Please describe in more details how you estimate ΔCO2 . What's the averaging interval? Do you filter the data in anyway. etc. This will also go to your methods section. Also it is better to bring the comparisons with other cities to the discussion/conclusion section.**

To our opinion, this comment made by the referee repeats in a certain way several previous comments relevant to description of the measurement campaigns and data processing algorithms. In fact, the answer to this comment of the referee has already been given above when we introduced appendixes and the subsections of the new Section 2 "Methods and instrumentation". In addition, in accordance with the referee's suggestion, we have moved the mentioned part of the text to Section 4 "Summary and conclusions".

**\*Line 155: what do you mean by geometry of the field experiment? You mean topography?**

For the sake of certainty, we have rephrased the text as follows:

It should be noted that the value of $\Delta XCO_2$ depends not only on the integral emission of the source, but also on its type (point, linear or area), the geometry of the field experiment (location of observations relative to the pollution plume) and on the meteorological situation during the measurements.

**\*Line 161: you mention here that the resolution of ODIAC is 1 km by 1 km. Later in the text you mention 0.93 by 0.46 km. This might confuse the audience please clarify.**

For clarity, we have rewritten this part of the text:

The nominal latitude/longitude size of the ODIAC data pixel is 30 arcseconds (Oda and Maksyutov, 2011), which defines a global spatial resolution of about 1 km × 1 km. Since the length of a degree of longitude changes with the latitude, the pixel size for St. Petersburg (~60° N) is smaller and equals to 0.93 km × 0.46 km (0.43 km$^2$).

**\*Line 213: Is this ODIAC value after you removed the pixel with 7000 kt/km2? If that's the case as mentioned above please clearly indicate it's not the original ODIAC value. If we add 7000 to this value then the model-measurement agreement might improve.**

No, this value (32529 kt) is an estimate of $CO_2$ annual emission coming from the original ODIAC database (see Section 2.2 "A priori data on FF $CO_2$ emissions"). If we smooth out the ejected pixel, as explained in the paper, we get 29524 kt. Thus, the difference is less than 10%, and this can hardly explain the observed discrepancy between the model and the measurements.

**\*Line 226: what is the time period of field observations?**

We now have this information in the new tables, 1 and 2 (see above). So, for clarity, we add the reference to one of these tables here:

For the sake of comparison, the simulation results and measurement data were averaged over time periods of field observations (the duration of each experiment is given in Table 1).

**\*Line 236: What is one observational series?**

In the revised version of the manuscript full details of both field campaigns are given in Tables 1 and 2 for 2019 and 2020, respectively. In addition, we provide an appendix showing the data series of measured and calculated $CO_2$ content for the selected campaign days in 2019 and 2020 – Appendix B, Fig. B1 and B2 (see above). Thus, we refer the reader to these plots in the appendix:

> The error bars indicated in Fig. 5a as boxes are in fact the variations of $\Delta CO_2$ obtained as standard deviation of observations and simulations within one observational series (see Appendix B, Fig. B1).

**\*Line 251. The sentence is unclear. Please consider rewording.**

The sentence was reworded as follows:

> An earlier analysis of the results of the EMME-2019 measurement campaign focused in particular on inferring the area fluxes of urban $CO_2$ emissions from St. Petersburg. In order to achieve this goal, the simplified mass balance approach was applied to the observed $CO_2$ enhancements ($\Delta CO_2$) which were attributed to the accumulation of pollution during the air mass movement on its way from the upwind side to the downwind side of the megacity (see Makarova et al., 2021 for full details).

**\*Line 294: The sentence is unclear. Please consider rewording or adding more explanation.**

The sentence was reworded as follows:

> Our confident expectation to detect the transport contribution is based on the high sensitivity of FTIR measurements of $XCO_2$ using EM27/SUN spectrometers and COCCON methodology.

**\*Line 302: Please specify in detail which days were excluded from the analysis and how many days are used after the exclusion. As mentioned earlier a table would be helpful.**

We add the following explanation to the text here:

> In general, for these reasons (including unfavorable weather conditions and wrong location of FTIR measurement points), data from only a few experiments were excluded: No.8 on April 18, 2019, No.10 on April 25, 2019, No.11 on April 30, 2019 (see Table 1) and No.4 on March 27, 2020 (see Table 2).

We believe that having Tables 1 and 2, as well as the figures in Appendices A and B, a reader has now complete information about the days and time, and the location of all our observations.

**\*Line 359: Please explain in more details why that's the case. From what I understand the in-situ measurement site is in a fairly remote location but the FTIRs have been deployed in the city centre and closer to large sources. So there are other variables that might play a role here!**

Let us explain here. Not only the in situ measurements in Peterhof, but also all mobile observations were planned in such a way that the FTIR measurement points were located at a distance from any local sources of heavy pollution. The main idea was to detect the integral $CO_2$ enhancement as the

difference between downwind and upwind observations. In this sense the Peterhof site may also appear inside the urban plume and get the high levels of $CO_2$ surface concentration. However, these surface $CO_2$ enhancements obviously lack the contributions of elevated emission sources being well mixed in the upper altitude layers. This is not the case for FTIR measurements which probe the column amount of $CO_2$ and not only the surface layer. We believe that the revised version of our manuscript with new plots and maps makes this clear and needs no further explanation.

**\*Figure 1 and 9: Please add latitude and longitude coordinates to the map.**

Coordinates have been added to the map.

**\*Figure 3: Please add latitude and longitude coordinates to the bottom map.**

Coordinates have been added to the bottom map.

**\*Figure 6: Please add latitude and longitude coordinates to the map. Also please specify the location of upwind instrument for 2019. It would also be helpful to specify the dates at each location.**

Coordinates have been added to the maps in Figure 6 (Figure 7 in the revised version of the manuscript). In order not to overload this figure with information, details about the location of the instruments for each day are presented in separate figures provided in the Appendix A.

**\*Figure 7: Are the ODIAC area flux values shown the average over the entire domain or the average along the HYSPLIT back-trajectory?**

The ODIAC area flux values are the average along the trajectories shown on the map in Figure 6 (Figure 7 in the revised version of the manuscript). We explain this in the text:

*Schematically, the air trajectories corresponding to the 2019 FTIR measurement locations are shown in Fig. 6. These trajectories were simulated as backward trajectories by the HYSPLIT model in the boundary layer of the atmosphere. The resulting values of anthropogenic $CO_2$ area fluxes calculated by integrating the ODIAC data along the trajectories presented in Fig. 6, are shown in Fig. 7 in comparison with the experimental estimates by Makarova et al., 2021. As in the study by Makarova et al., 2021, the width of the air paths was assumed to be 10 km*

We also specify this in the caption to Figure 7 (Figure 8 in the revised version of the manuscript):

*Figure 7: The $CO_2$ area flux ($F_{CO2}$) obtained on the basis of the mass balance approach (EMME-2019) compared to the $CO_2$ area flux derived from scaled ODIAC data. The calculations are made for the trajectories shown in Fig. 6. Dots are connected by lines for illustrative purposes only.*

---

## Author Comment (AC2)

Authors' reply to referee #2

We thank the referee for reviewing our paper and for the critical remarks. This helped us to realize the shortcomings in the structure of our work and to reveal missing data and useful illustrative materials. We have partially reorganized the paper and supplied it with additional tables, graphs, and maps. We believe that all these changes and additions will make our study clearer for understanding and will successfully address all the issues raised by the reviewer.

Below, the actual comments of the referee are given in **`bold courier font and blue colour`**.
The text added to the revised version of the manuscript is marked by red colour.

**`*Although the amount of data used is quite limited (11 days using 2 instruments in 2019, 6 days with one instrument in 2020), the authors hope to build on previous studies that have shown the utility of groups of EM27/SUN sensors to detect small enhancements in trace gas column concentrations associated with urban emissions. Their ultimate goal is to use the little data they have to determine the emission rate of the entire city. Anyone who has attempted to infer urban emissions from scarce atmospheric observations of this kind will recognize the difficulty of this task, as there are numerous sources of noise and uncertainty that are hard to account for.`**

We agree with the referee and recognize the difficulty of our task. However, in our research, we rely on measurements performed by high-precision and well-calibrated spectral equipment – EM27/SUN FTIR spectrometers, which have been successfully used in several similar field campaigns. Our team of researchers has extensive experience in estimating anthropogenic emissions from experiments, including those based on various remote measurements in the area of St. Petersburg. Thus, EMME campaigns of 2019 and 2020, the results of which we use in this work, are definitely not our first experiments of this kind. Below we present several references to our previous studies:

Ionov, D.V. and Poberovskii, A.V.: Quantification of $NO_x$ emission from St. Petersburg (Russia) using mobile DOAS measurements around entire city, Int. J. Remote Sensing, 36, 2486-2502, https://doi.org/10.1080/01431161.2015.1042123, 2015.

Ionov, D.V. and Poberovskii A.V.: Integral emission of nitrogen oxides from the territory of St. Petersburg based on the data of mobile measurements and numerical simulation results, Izv. Atmos. Ocean. Phys., 53, 204-212, https://doi.org/10.1134/S0001433817020049, 2017.

Makarova, M.V., Arabadzhyan, D.K., Foka, S.C., Paramonova, N.N., Poberovskii, A.V., Timofeev, Yu.M., Pankratova, N.V., and Rakitin, V.S.,: Estimation of nocturnal area fluxes of carbon cycle gases in Saint Petersburg suburbs, Russ. Meteorol. Hydrol., 43, 449-455, https://doi.org/10.3103/S106837391807004X., 2018.

We would like to emphasize that although the field measurements of our study are limited in time, they form a set of a series of data that is not scarce but quite typical for an intensive measurement campaign and cover a period from March to May for two adjacent years, 2019 and 2020. It is indeed common that atmospheric observations of this type are few in number, as they are very resource-intensive and time-consuming, and highly depend on favorable weather conditions.

We agree with the referee that installation of a permanent city observation network based on several EM27/SUN units would be a desirable ultimate goal. We hope that this paper describing the successful

pilot study will help to raise the required funding for extending the collaboration between Russian and EU researchers fostering the quantification of greenhouse gas emissions in Russia.

**\*This paper falls short, however, on presenting a convincing method for retrieving an urban emission rate using this data. Most strikingly, the entire manuscript lacks detailed equations describing exactly how the FTIR data, transport model, and ODIAC data, and combined.**

We agree with this comment. As we noted above, the paper has been significantly revised and provided with additional material (equations, tables, plots), making our results more convincing. In particular, the equation mentioned by the reviewer was added to the text (subsection 3.1 "The results of the EMME-2019 campaign"):

In order to obtain a quantitative agreement between simulated and observed $\Delta CO_2$, the inventory data (the ODIAC emissions), which are used as input information for the HYSPLIT dispersion model, should be scaled (Flesch et al., 2004). The scaling factor ($SF$) is derived as follows. The data from all days of measurements are compared to the corresponding model simulations, see Fig.6 as an example of a scatter plot. The scaling factor is determined as a slope value of the following regression line (e.g. the slope is $2.88 \pm 0.21$ , as shown in Fig.6):

$$\Delta CO_2[FTIR]_i = SF \times \Delta CO_2[HYSPLIT_{ODIAC}]_i \qquad (1)$$

where $\Delta CO_2[FTIR]_i$ is is the difference between the downwind and upwind FTIR measurements averaged over the duration of experiment $i$ (see Table 1 and Table 2, Appendix A and Appendix B for the details of every field experiment) and $\Delta CO_2[HYSPLIT_{ODIAC}]_i$ is the averaged difference between the downwind and upwind $CO_2$ column calculated using the HYSPLIT dispersion model for the location and time of FTIR observations, and initialized with ODIAC $CO_2$ emissions.

**\*I would expect a paper that presents a data-model fusion product like this to not only have extensive equations and tables, but also a supplement with additional tables and figures, but there appears to not be any supplemental information provided.**

We agree with this comment. Apparently, it was our mistake to refer too much to our previous study (Makarova et al., 2021), so as not to reproduce the same material in both papers. Following the reviewer's suggestion, we fix this and supply a set of additional figures available in the Appendix A and B of revised manuscript. Please, see Appendix A below:

[revised manuscript text omitted]

*Even the EM27 data itself is not presented clearly in this manuscript. It would be useful to see a couple of daily time series plots of the XCO$_2$ data from both sensors so the reader can see not only the difference between them, but the (likely) large hourly variations typically seen by urban EM27 instruments.

Addressing this issue, we have provided the revised version of the manuscript with an appendix showing the data series of measured and calculated CO$_2$ content for the selected campaign days in 2019 and 2020. The graphs in the figures clearly demonstrate that the "downwind-upwind" CO$_2$ enhancements are reliably detected with FTIR measurements during each mobile experiment and are not masked by the hourly CO$_2$ variations. Please, see Appendix B below:

**Appendix B: The data series of measured and calculated CO$_2$ content**

The upwind and downwind CO$_2$ total column values acquired from FTIR measurements and HYSPLIT calculations are shown for selected campaign days in 2019 and 2020 in Fig. B1 and B2. The HYSPLIT data are in fact the values of an integrated vertical column in the range of 0-1500 meters (10 altitude layers) calculated with the 15-minute time step. The background level of the CO$_2$ column is set equal to an average of the FTIR upwind measurements during a day. Note that in 2020, there were days when the downwind measurements were performed twice, at different locations – on March 22 and May 1 (see Fig. B2).

[Figure]

**Figure B1: Time series of measured (FTS) and simulated (HYSPLIT, without scaling of the ODIAC emissions data) CO$_2$ total column at the upwind (blue lines) and downwind (red lines) locations for selected campaign days in 2019.**

[Figure]

**Figure B2: Time series of measured (FTS) and simulated (HYSPLIT, without scaling of the ODIAC emissions data) $CO_2$ total column at the upwind (blue lines) and downwind (red lines) locations for selected campaign days in 2020.**

In addition, two examples of HYSPLIT-simulated $CO_2$ plumes and corresponding time series of $CO_2$ total column measurements for the typical days of experiments in 2019 and 2020 are presented in Figs. 5 and 9:

[Figure]

**Figure 5: Left panel: Urban pollution $CO_2$ plume over St. Petersburg calculated by HYSPLIT model for April 4, 2019 (10:00 UTC). The colour bar designates the $CO_2$ total column in units $10^{21}$ cm$^{-2}$. The blue and red circles indicate the locations of upwind and downwind FTS observations, accordingly. Right panel: Time series of measured (FTS) and simulated (HYSPLIT, without scaling of the ODIAC emissions data) $CO_2$ total column at the upwind (blue lines) and downwind (red lines) locations for the same day.**

[Figure]

**Figure 9: Left panel: Urban pollution $CO_2$ plume over St. Petersburg calculated by HYSPLIT model for April 8, 2020 (10:00 UTC). The colour bar designates the $CO_2$ total column in units $10^{21}$ cm$^{-2}$. The blue and red circles indicate the locations of upwind and downwind FTS observations, accordingly. Right panel: Time series of measured (FTS) and simulated (HYSPLIT, without scaling of the ODIAC emissions data) $CO_2$ total column at the upwind (blue lines) and downwind (red lines) locations for the same day.**

*It is not entirely clear how background $XCO_2$ concentrations are determined. For the 2019 campaign, when 2 FTIRs were used, it appears that the sensors were placed such that one was inside the "urban plume" and that one was placed outside of this plume. The sensor outside the plume is then assumed to be the background, but there is nothing presented in this manuscript that builds confidence that this a reasonable assumption. Is the background site even upwind of the city? What is the uncertainty associated with this decision? Are there emission sources upwind of this background site? For the 2020 data, the background determination is even worse, as only one instrument was available, so the sensor was moved during the course of the day in an attempt to capture a useful background value. Unfortunately, total-column $CO_2$ concentrations can vary greatly over the course of a day, and it is not uncommon for background variations to be on the order of a urban emissions signal, making this assumption unadvisable.

In part, the answer to the reviewer's questions is contained in the figures mentioned above - Figure 5 and 9, Figure A1 and A2 (Appendix A) and Figure B1 and B2 (Appendix B). The HYSPLIT simulation maps, based on the ODIAC $CO_2$ emission inventory, clearly show that the "upwind" FTS instruments are placed indeed on the upwind side of the city, in the background area, with no sources of emission upwind of the background site. Besides, the data series of FTS measurements at the background site show mainly stable $CO_2$ behavior. We would like to emphasize that in any case variations of the background $CO_2$ are considerably lower than an urban emissions signal.

*The implementation of the transport model is also questionable. The authors state that they are using the HYSPLIT dispersion model, but nowhere in the figures or texts does it appear that any dispersion is actually being simulated. It is unclear, but it looks like HYSPLIT was configured to run backwards in time to compute single particle trajectories, with no stochastic (dispersion) component. It is then stated that "The width of the air paths was assumed to be 10km" [Line 262], which I assume means that plume of influence on each observation is simply modelled as a straight line 10km wide. This type of modelling would suggest that the column observed is equally sensitive to emissions 500 meters upwind as it is to emissions 15 km upwind, which is incorrect. It is then unclear how surface emissions are "integrated" into the column based on these trajectories. Also, how is vertical transport dealt with? Are particles that rise to the top of the boundary layer treated the same as those that travel closer to the surface?

Unfortunately, the reviewer got the wrong idea about this part of our work. Perhaps this is due to the insufficient amount of illustrative material in the original version of our manuscript. In the revised version we tried to do our best in order to present the method and the details of modeling as clear as possible. However, we would like to note that in fact, the scheme of using the HYSPLIT model was outlined by us already in the original version (see subsection 3.2 "HYSPLIT model general setup"):

> *The spatial and temporal evolution of the urban pollution plume was simulated using the HYSPLIT model (Draxler and Hess, 1998; Stein et al., 2015). Calculations were performed for the territory of the St. Petersburg agglomeration using the offline version of the HYSPLIT model with the setup similar to the one that was successfully used previously for the $NO_x$ plume modelling (Ionov and Poberovskii, 2019; Makarova et al., 2021). A 3-dimensional field of anthropogenic air pollution was calculated for a spatial domain with coordinates 54.8°-61.6° N, 23.7°-37.8° E; the domain grid size is 0.05°×0.05° latitude and longitude (see Fig. 3, top). The vertical grid of the model is set to 10 layers with the altitude of the upper level at 1, 25, 50, 100, 150, 250, 350, 500, 1000 and 1500 meters a.s.l., respectively. As a source of meteorological information (vertical profiles of the horizontal and vertical wind components, temperature and pressure profiles, etc.), the NCEP GDAS (National Centers for Environmental Prediction Global*

*Forecast System) data were used, presented on a global spatial grid of 0.5° × 0.5° latitude and longitude with time interval of 3 hours (NCEP GDAS, 2020).*

We now add some more details of the HYSPLIT configuration in the revised version of our manuscript:

To run HYSPLIT we used the software package: HYSPLIT 4, June 2015 release, subversion 761. The advanced setup of the HYSPLIT model was configured as follows (basic parameters):
- default method of vertical turbulence computation,
- horizontal mixing computed proportional to vertical mixing,
- boundary layer stability computed from turbulent fluxes (heat and momentum),
- vertical mixing profile set variable with height in the planetary boundary layer (PBL),
- boundary layer depth set from the meteorological model (input meteorology data),
- puff mode dispersion computation with a "tophat" concentration distribution on a horizontal and vertical scale.

We believe that taking into account new illustrative materials added to the revised manuscript, the reader will get a complete view of the intensive model calculations which we carried out. We have also to mention, that quite similar HYSPLIT simulations helped us to plan each of the field experiment in 2019 and 2020. For this purpose, the forecasts of urban plume evolution have been calculated. To get an idea of the degree of spatio-temporal detail of these calculations, the reviewer can look at animations, available from the following direct links:

| | |
|---|---|
| 21 March 2019: | https://youtu.be/0GD8_YsNt2Q |
| 27 March 2019: | https://youtu.be/40mGPgkCAmw |
| 01 April 2019: | https://youtu.be/Gc3LUV4jmVI |
| 03 April 2019: | https://youtu.be/cPZ-71ZvKHw |
| 04 April 2019: | https://youtu.be/ekc2ip9OplY |
| 06 April 2019: | https://youtu.be/rgtq6JLPhig |
| 16 April 2019: | https://youtu.be/1POH1GghvXA |
| 18 April 2019: | https://youtu.be/PByNmoR800E |
| 24 April 2019: | https://youtu.be/jBydWV84XQY |
| 25 April 2019: | https://youtu.be/fFwb-AuitxU |
| 30 April 2019: | https://youtu.be/9y7SC29iEgI |
| 22 March 2020: | https://youtu.be/5-fyy69DdV4 |
| 23 March 2020: | https://youtu.be/KZD6c23BDDY |
| 27 March 2020: | https://youtu.be/p2vx3RyAq0U |
| 05 April 2020: | https://youtu.be/kf7YAI1PFyg |
| 08 April 2020: | https://youtu.be/xcQyDO8IjbA |
| 01 May 2020: | https://youtu.be/GlicSVAZIyU |

One should keep in mind that the animations presented above are not directly related to the calculations relevant to the present study, since they model the evolution of the tropospheric $NO_x$ plume in time increments of one hour, using the forecast meteorology data (for full details see Makarova et al., 2021). Our HYSPLIT simulations of $CO_2$ are even more detailed, as they utilize the ODIAC emissions inventory and reanalysis meteorology data to calculate $CO_2$ concentrations with 15-minute time step.

**\*The current version of HYSPLIT is able to run in a mode that actually simulates dispersion and surface influence on observations, using the Stochastic Time-Inverted Lagrangian Transport (STILT) model. The HYPSLIT-STILT model produces a**

influence function (footprint) with the correct units ( ppm / umol/m2s ) to relate
surface emissions to atmospheric observations, and have been used many times in
studies with similar goals as this one. I would strongly suggest using this, or a
similar model, to reprocess these results.

We hope that all our answers to the referee's comments and the new plots shown in the revised version
will definitely convince the reviewer that in this paper we have used the capabilities of dispersion
modeling with HYSPLIT tools in their entirety. To our opinion, the functions of the HYSPLIT model
are quite sufficient for the tasks we solve, which is confirmed, in particular, by our own long-term
experience in HYSPLIT atmospheric modeling. One can mention, for example:

Ionov, D.V. and Poberovskii, A.V.: Nitrogen dioxide in the air basin of St. Petersburg: Remote
measurements and numerical simulation, Izv. Atmos. Ocean. Phys., 48, 373–383,
https://doi.org/10.1134/S0001433812040093, 2012.

Ionov, D.V. and Poberovskii, A.V.: Quantification of $NO_x$ emission from St. Petersburg (Russia) using
mobile DOAS measurements around entire city, Int. J. Remote Sensing, 36, 2486-2502,
https://doi.org/10.1080/01431161.2015.1042123, 2015.

Ionov, D.V. and Poberovskii A.V.: Integral emission of nitrogen oxides from the territory of St.
Petersburg based on the data of mobile measurements and numerical simulation results, Izv.
Atmos. Ocean. Phys., 53, 204-212, https://doi.org/10.1134/S0001433817020049, 2017.

Ionov, D. V. and Poberovskii A. V.: Observations of urban $NO_x$ plume dispersion using the mobile and
satellite DOAS measurements around the megacity of St. Petersburg (Russia), Int. J. Remote
Sens., 40, 719-733, https://doi.org/10.1080/01431161.2018.1519274, 2019.

We do not see any need to involve the HYPSLIT-STILT model to this study, since our HYSPLIT
calculations already take into account all the effects mentioned by the reviewer. It is also worth noting
that all the calculated fields of the $CO_2$ content are already presented in the correct units – [molecules
cm$^{-2}$] for total column and [ppmv] for concentration (one can make sure of this by looking at Figure 4
and Figure X1 and X2 of the manuscript).

*It is unclear (due to the lack of math presented) how the observations, transport
model, and prior inventory are combined to produce the resulting emissions scaling
factors and uncertainties. Did the fitting process take into consideration
different uncertainties in the model and observations? It is mentioned that "The
error assessment for the scaling factor should be discussed is some detail" [Line
231], however this is followed by only a few sentences which present an error
analysis that does not account for any large sources of error, such as errors in
the transport do to wind speed and direction uncertainty, or errors due to
uncertainties in the background estimate or spatial distribution of emissions.

It seems to us that taking into account a number of additions made to the text and new illustrative
material, the revised manuscript already contains answers to the questions raised here by the reviewer.
In particular, the full use of the HYSPLIT dispersion modeling tools minimizes errors associated with
wind direction and speed uncertainty. Meteorological data that is used as input to the HYSPLIT
simulation pass a rigorous preprocessing to ensure its self-consistency. Getting information on spatial
distribution of emission sources is a complicated task itself, especially if emissions are considered for
the megacity at high resolution grid. Therefore we use in our study the recognized and open access
ODIAC database. Any varying of the position/intensity/type of emission sources should have well-
reasoned basis.

We also slightly edited the part of the text ("*The error assessment for the scaling factor ..*") that the reviewer pointed out in his comment:

The error of the scaling factor was estimated under the assumption that the measurement errors are the same for all days as well as the model simulation errors. The error bars indicated in Fig. 6 as boxes are in fact the variations of $\Delta CO_2$ obtained as standard deviation of observations and simulations within one observational series (see Appendix B, Fig. B1). Obviously, these quantities comprise both measurement errors and simulation errors (including those associated with wind direction and speed uncertainty), and temporal variability of the $CO_2$ TC. One can see that these quantities differ from day to day.

***It is my opinion that the work as is does not present a robust, reproducible, or innovative analysis that adds scientific value to the dataset.**

We completely disagree with the reviewer's last comment. In contrast to his/her opinion, we would like to note the following:

1. Robustness.

Our study is based on the measurements made by state-of-the-art spectroscopic instruments, namely EM27/SUN FTIR spectrometers, during two consecutive field campaigns. An EM27/SUN spectrometer has already proven itself in the scientific community as an appropriate tool for studies of the horizontal inhomogeneities of atmospheric composition on a regional scale. For simulations of the 3D $CO_2$ field, we used the HYSPLIT model in the dispersion calculation mode. HYSPLIT is a well known and widely used tool for the simulation of this kind. The approach based on emission scaling is also well known and has proven its efficiency. The a priori emissions database ODIAC is also widely used. So, we do not see any reasons to consider our analysis as not robust.

2. Reproducibility.

If necessary, every number and plot presented in our paper can be easily reproduced by any researcher. As indicated in the "Data availability" section of the manuscript: "The datasets containing the EM27/SUN measurements during EMME-2019 and EMME 2020 can be provided upon request". Then, the HYSPLIT model package is available for free download from the NOAA Air Resources Laboratory at https://www.ready.noaa.gov/HYSPLIT.php. For example, to produce a 3-day 3D simulation of 15-minutes step $CO_2$ concentration over the spatial domain of our interest it takes only one hour of processing time by standard Microsoft Windows XP personal computer (Intel Core i3-4150 CPU @ 3.50Ghz, 4GB RAM). Anyone who has attempted to run such atmospheric simulations will recognize that it is really cost-effective processing setup.

3. Innovative character.

The results of our research provide an independent and new top-down estimate of $CO_2$ emission by one of the main industrial cities of European Russia. This estimate is approximately twice the value indicated in modern global emission inventories (such as, for example, the well known ODIAC database).

We guess that the referee's opinion expressed in the last comment is based mainly on the lack of detailed information on FTIR measurements and HYSPLIT simulations in the original manuscript. We do hope that our extensive revision of the article will make the presentation of our study more clear. And we thank the referee once again for the insightful and helpful comments.

---

## Author Comment (AC3)

Authors' reply to Dr. Timofeyev

We are grateful to Dr. Timofeyev for his interest in our paper.

Below, the actual comments of the referee are given in **`bold courier font and blue colour`**.
The text added to the revised version of the manuscript is marked by red colour.

**`*The problem investigated in the study is important and relevant due to the Earth climate change and the importance of megacities for the variation of the atmospheric gas composition. Therefore, the authors should be welcomed to keep providing studies on the independent assessment of such emissions.`**

We thank Dr. Timofeyev for this positive assessment of our work.

**`*The different estimates of the St.Petersburg integral emissions which are in range from 44800 to 74800 kt/year are given in the article. The difference between the minimum and maximum of the emissions constitutes approximately 31000 kt/year or ~70% relatively to the minimal value. The variations have to be analyzed, the inaccuracies of the approaches applied and natural variations have to be assessed. What is the reason for such a big spread between emission estimates - the technique of the measurements, lack of the observation data or their quality, the natural emission variation, the influence of the different trajectories, etc? The analysis of the estimated emissions and their uncertainties (random and systematic), the measurement technique and the inversion modelling approach used in the study have to be provided in the article.`**

Actually, our study reveals 75800±5400 kt/year from the field measurements in 2019 and 68400±7100 kt/year from the field measurements in 2020. Thus the difference between these two is just 10%, so the "70% difference" is definitely out of the question here. Apparently, the 10% difference between the two estimates in 2019 and 2020 is rather small and looks quite reasonable. As for the value of 44800 kt/year – this is an estimate based on the analysis of the ground-level measurements of $CO_2$ surface concentrations, carried out by gas analyzer at the site of Peterhof (see the subsection 3.3 "Simulations of ground-level $CO_2$ concentrations" of the original manuscript for full details). This estimate stands aside, and the reasons for this are discussed in the original version of our manuscript (subsection 4.1 "The results of the EMME-2019 campaign"):

> *Resulting $CO_2$ emission rate is almost twice as high as the above estimate, based on the analysis of ground-level $CO_2$ measurement data (Section 3.3, 44800±1900 kt year$^{-1}$). This difference may indicate a significant contribution of elevated $CO_2$ sources (industrial chimneys) that could not be registered by the ground-level in situ measurements, as the elevated exhausts of pollution are more likely to further rise up, rather than descend to the ground. In contrast, FTIR measurements of the total column keep being sensitive to this kind of emissions. In addition, while FTIR measurements implement a "cross section" of the urban pollution emission zone in a series of multidirectional trajectories (depending on the wind direction), local ground-level in situ measurements at a specific location (Peterhof) can not capture the contribution of the entire mass of urban emissions. Thus, estimates of integral $CO_2$ emissions based on the interpretation of ground-level measurements in Peterhof can be considered as a lower limit of an estimate.*

**`*The significant systematic errors of the integral emission estimation approach used in the study can be related to the trajectories applied in the approach. The analysis of the Fig.6 demonstrates that the trajectories which link the positions of the observations cover the city irregularly. For instance, there are large city`s areas which were not covered by the trajectories completely. By contrast,`**

**some of the city's zones were covered by the measurements (which after that were used in the emission estimation) several times.**

First, we would like to emphasize that there are no "significant systematic errors of the integral emission estimation approach" reported in our paper (see the comment above). On the contrary, the differences in estimates are quite insignificant. Second, the air mass trajectories shown in Fig.6 relate to the problem of determining the area fluxes by the mass balance approach adopted in the form of a one-box model. This part of the paper is a very small element of our study, and it is intended to demonstrate the agreement with similar results presented earlier by Makarova et al. (2021). This section has nothing to do with the main task of determining the integral $CO_2$ emission based on the comparison of FTIR measurements with HYSPLIT simulation data.

**\*Since the quality of a priori information (especially the accuracy of a transport model) is crucial for the quality of inverse modelling, readers can be interested by the comparison of the local measurements of $CO_2$ mixing ratio in Peterhof and HYSPLIT modelled data. The quantitative analysis (STD, MAE, RMSE) of such comparison before and after the scaling of the a priori emissions have to be provided in the study.**

This is exactly what we did and what is already available in the original version of the manuscript. We have the impression that Dr. Timofeyev missed section 3.3 "Simulations of ground-level $CO_2$ concentrations":

*Routine measurements of $CO_2$ surface concentrations have been carried out at the atmospheric monitoring station of St. Petersburg University in Peterhof (59.88° N, 29.82° E) since 2013. These observations are the in situ measurements using a gas analyzer Los Gatos Research GGA 24r-EP. The instrument is installed on the outskirts of a small town of Peterhof in the suburbs of St. Petersburg (see location in Fig. 1). This place is far enough away from busy streets and other local sources of pollution, with an ambient air intake being 3 meters above the surface. To test the HYSPLIT model setup for the St. Petersburg region, we calculated the surface concentration of $CO_2$ near the Peterhof during the 2019 EMME measurement campaign – from March 20 to April 30, 2019 (Makarova et al., 2021). The results of the model calculations were compared to the data of in situ measurements (due to the instrument failure in 2020 the comparison is limited to the period of EMME campaign in 2019 only). Observational data and simulation results were averaged over 3-hour intervals. The resulting comparison is shown in Fig. 4. The model reproduces the temporal variations of $CO_2$ including the main periods of significant growth of concentration; the correlation coefficient between the calculation and measurements is equal to 0.72. The background value of the surface concentration is taken as 415 ppmv based on long-term local measurements. It is important to emphasize that quantitative agreement is achieved by linear scaling of the a priori integral urban $CO_2$ emission. The scaling coefficient for emissions corresponds to the value of the integral urban $CO_2$ emission from the territory of St. Petersburg of $44800\pm1900$ kt year$^{-1}$ (the given uncertainty is due to the uncertainty of the fitted scaling factor). This value is noticeably higher than official estimates mentioned above and ODIAC data for 2018 (32529 kt). The average discrepancy between the measurement and simulation data shown in Fig. 4 is $2\pm9$ ppmv (model calculations are systematically lower).*

[Figure]

*Figure 4: Comparison of the HYSPLIT simulations and the in situ measurements of surface $CO_2$ concentration in Peterhof (59.88° N, 29.82° E) in March-April 2019. Left panel: The values of surface $CO_2$ compared with the results of HYSPLIT simulations before scaling of the ODIAC emissions data. Right panel: HYSPLIT data obtained using scaled ODIAC $CO_2$ emissions compared with observed surface $CO_2$. Measurement and simulation data are averaged over 3-hour intervals.*

**\*The authors give insufficient review on the $CO_2$ and other greenhouse gases emission estimates provided for Moscow and St.Petersburg megacities by other researchers.**

To our knowledge, there are no other studies of the $CO_2$ emissions either in St. Petersburg or in Moscow megacities published in the peer-reviewed scientific literature and relevant to the specific topics of our research. Dr. Timofeyev in the beginning of his comment mentions the paper by Y.M. Timofeyev, G.M. Nerobelov, Y.A. Virolainen, A.V. Poberovskii, and S.C. Foka: "Estimates of $CO_2$ anthropogenic emission from the megacity St. Petersburg", Dokl. Earth Sc. 494, 753-756 (2020), https://doi.org/10.1134/S1028334X20090184. We are certainly aware of this work, since the two of its co-authors – Y.A. Virolainen and S.C. Foka – are active participants of both 2019 and 2020 EMME campaigns and they are the co-authors of our present paper. The mentioned study by Dr. Timofeyev et al. (2020) basically exploits the experience of EMME-2019 campaign and confirms one of the main of its findings – the twofold underestimation by the official inventory of the $CO_2$ emission. This finding had been already reported in the discussion paper submitted by Makarova et al. to AMT in April 2020 and later on accepted for publication and published in the beginning of 2021. Dr. Timofeyev et al. in their work used the data of observations performed within EMME-2019 and combined it with an ODIAC emissions inventory. It should be emphasized that according to the tradition of the Russian Academy of Sciences articles are submitted to the Journal "Dokl. Earth Sc." only with a recommendation from a member or corresponding member of the Academy and these articles do not go through a standard peer-review process (the article by Timofeyev et al. (2020) was received June 16, 2020; revised June 17, 2020; accepted June 18, 2020). To our opinion, the absence of any peer review has led to the fact that this paper may contain inaccuracies. For example, a reader can come to the conclusion that the authors used an extremely simplified assumption for the transfer of air masses: strictly straight from the measurement point on the upwind side to the measurement point on the downwind side (see Fig. 1 on page 754, Timofeyev et al., 2020). If so, then this approach is rather questionable, since in reality the wind direction has never exactly coincided with the straight line connecting these two measurement points. We would like to emphasize that the article by Timofeyev et al. (2020) is extremely short and is missing a lot of information which is important. For example, the authors mention the tomographic approach to the analysis of measurement data by, but do not reveal the essence of this method at all. It should be noted that the authors did not acknowledge the owners of the used equipment (FTIR spectrometers EM27/SUN from Karlsruhe Institute of Technology,

Germany) and the contribution of the German participants of the EMME experiment. Timofeyev et al. (2020) did not specify the sources of funding for the measurements campaign (European Union's Horizon 2020 research and innovation programme under grant agreement No 776810, VERIFY project; Russian Foundation for Basic Research through the project No.18-05-00011). To the first glance, the absence of this information may seem a formal fault, however, to our opinion, it can mislead readers and produce a wrong impression about the EMME experiment and the priorities of the obtained results. Therefore, we decided to avoid citing the mentioned article by Timofeyev et al. (2020) in our present paper.

**\*A descriptive table containing details of the 2020 measurement campaign (e.g. atmospheric conditions with its dynamic, etc) has to be added to the article how it was done in the previous study.**

We supplied Section 2 "Methods and instrumentation" of the revised manuscript with the tables of information about each mobile experiment in 2019 and 2020 (see below):

[revised manuscript text omitted]